# Modelling steady states and the transient response of debris-covered glaciers

James C. Ferguson[1] and Andreas Vieli[1]

[1]Glaciology and Geomorphodynamics Group, Institute of Geography, University of Zurich

**Correspondence:** James C. Ferguson (james.ferguson@geo.uzh.ch)

**Abstract.** Debris-covered glaciers are commonly found in alpine landscapes of high relief and play an increasingly important role in a warming climate. As a result of the insulating effect of supraglacial debris, their response to changes in climate is less direct and their dynamic behaviour more complex than for debris-free glaciers. Due to a lack of observations, here we use numerical modelling to explore the dynamic interactions between debris cover and geometry evolution for an idealized glacier over centennial timescales. The main goal of this study is to understand the effects of debris cover on the glacier's transient response. To do so, we use a numerical model that couples ice flow, debris transport and its insulating effect on surface mass balance and thereby captures dynamic feedbacks that affect the volume and length evolution. In a second step we incorporate the effects of cryokarst features such as ice cliffs and supraglacial ponds on the dynamical behaviour. Our modelling indicates that thick debris cover delays both the volume response and especially the length response to a warming climate signal. Including debris dynamics therefore results in glaciers with extended debris-covered tongues and that tend to advance or stagnate in length in response to a fluctuating climate at century time scales and hence remember the cold periods more than the warm. However, when including even a relatively small amount of melt enhancing cryokarst features in the model, the length is more responsive to periods of warming and results in substantial mass loss and thinning on debris covered tongues, as is also observed.

## 1 Introduction

Debris-covered glaciers are commonly found in alpine landscapes of high relief, often when a primary source of mass input to the glacier comes from avalanching. Steep headwalls and slopes deliver debris consisting of loose rocks onto the glacier surface, mixed in with ice and snow. Typically, this debris falls in the accumulation zone and becomes entrained in the ice, emerging on the surface further down-glacier in the ablation zone after it is left behind as the ice melts. Debris may also be delivered directly to the glacier surface when avalanching occurs in ablation zone.

A debris-covered glacier is commonly defined as any glacier with a continuous debris cover across its full width for some portion of the glacier (Kirkbride, 2011). For a thin layer of debris, the resulting decrease in surface albedo leads to an elevated melt rate of the underlying ice; however, when the debris cover exceeds a thickness of a few centimetres, it reduces the ablation of the underlying ice (Østrem, 1959; Nicholson and Benn, 2006). For highly debris-covered glaciers, this reduced melt

rate leads to glaciers with larger volumes and greater extents than would be expected for the corresponding debris-free case (Scherler et al., 2011).

Debris-covered glaciers exhibit a wide range of responses to changes in climate, some of which are counterintuitive (Scherler et al., 2011). Many debris-covered glaciers globally are retreating, particularly in the Himalayas (Bolch et al., 2012), though more slowly and with stagnant termini. Additionally, some debris-covered glaciers exhibit mass loss rates that are similar to those observed for nearby debris-free glaciers (Kääb et al., 2012; Gardelle et al., 2013; Pellicciotti et al., 2015; Brun et al., 2018) and that have been related to enhanced thinning rates on their debris covered glacier tongues. The formation of features such as ice cliffs and supraglacial ponds, which we refer to collectively here as cryokarst features, has been suggested as a potential explanation for this anomalous thinning and therefore the occurrence of such features and their enhancing effect on surface melt have been intensively studied (Sakai et al., 2000, 2002; Steiner et al., 2015; Buri et al., 2016; Miles et al., 2016). However, the influence of dynamic effects on the thinning rates and glacier evolution has so far largely been neglected and such dynamic effects remain poorly understood. Further, from the reduction in ablation on debris covered glaciers a more delayed and dampened response is expected (Benn et al., 2012). Ideally one would use observational data across a greater temporal and geographical spectrum so that process feedbacks can be observed and examined over relevant time scales. However, the lack of long-term remote sensing data means that we have severe constraints on the availability of such long observational time series and the recent reconstruction of Zmuttgletscher (Mölg et al., 2019) provides the only currently available observable record of a debris-covered glacier that goes beyond a century.

Given the paucity of longterm data it is therefore essential to use advanced numerical models in order to investigate the role of glacier dynamics on glacier evolution and mass loss, allowing for the study of interacting processes over longer time-frames. Recent progress with numerical simulations of debris-covered glaciers include: Konrad and Humphrey (2000), where an early steady-state model of debris-covered glaciers was developed; Vacco et al. (2010) and Menounos et al. (2013), both of which studied the effect of a rock avalanche on glacier evolution using a coupled debris transport and ice dynamics model; Banerjee and Shankar (2013), which suggested that the transient response of a debris-covered glacier to changes in climate has two distinct timescales; Rowan et al. (2015), which was the first modern model of coupled debris-ice dynamics to study the longterm evolution of a debris-covered glacier; and Wirbel et al. (2018), which tested both 2-D and 3-D advective debris transport using a full-Stokes solver for the ice dynamics. Perhaps the most significant modelling study to date is Anderson and Anderson (2016), where certain technical issues are addressed in detail for the first time, such as how to handle both the boundary condition at the glacier terminus and the possibility of a variable debris source in the accumulation area. The body of work that uses essentially the same model has examined diverse topics relating to the feedbacks that exist between debris flux, debris thickness patterns, and steady-state glacier extent; has also studied the transient relationship between debris-covered glaciers and rock glaciers; and has been used to explore the processes that govern the age of ice-cored moraines (Anderson and Anderson, 2016; Crump et al., 2017; Anderson and Anderson, 2018; Anderson et al., 2018).

However, to date no study has used a coupled ice flow-debris transport model to systematically and in detail study the transient response and characteristic response times of a debris-covered glacier. A better understanding of how a debris-covered glacier responds to changes in climate, and what role the debris concentration and the prevalence of cryokarst features play in determining the magnitude of this response, is critical to predicting how today's debris-covered glaciers will evolve as the Earth's climate changes. Therefore, this study aims to fill the gap by investigating the difference in transient response of debris-covered glaciers from their debris-free counterparts. In particular, we: (1) examine how debris cover changes the transient response of an idealized glacier to step changes in climate, quantifying both the volume and length response; (2) examine the response to a fluctuating climate signal on the long-term evolution of am idealized debris-covered glacier as a function of debris concentration; and (3) examine the impact on mass loss and surface evolution when cryokarst features are included, quantifying this impact as a function of cryokarst area.

## 2 Methods

### 2.1 Governing equations

In order to examine the essential features of the interaction between glacier dynamics and debris cover, we couple an ice flow model to a debris transport model that includes both the debris melt-out and its insulating effect on ice ablation. In this model, the debris evolution affects the geometry and ice flow through changes in the surface mass balance. Our model is similar to that used in Anderson and Anderson (2016) with the main differences being some simplifications in the description of the ice flow, no explicit englacial debris tracking within the ice, and the novel option of melt enhancement due to cryokarst features. This cryokarst component is coupled to the flow dynamics and switches on when the tongue becomes stagnant.

#### 2.1.1 Ice dynamics

For ice flow, we use a flowline version of the Shallow Ice Approximation (SIA), a simple model that allows for a realistic qualitative study of a glacier's response to changing climatic condition. The SIA has been used for studying glacier evolution and response times for debris-free glaciers (e.g., Leysinger Vieli and Gudmundsson, 2004), where it achieved comparable results to a full-Stokes solver with significantly less computational time. For a glacier with evolving ice thickness $H(x,t)$ flowing along the down-glacier direction $x$ with depth averaged velocity $\bar{u}(x,t)$ in response to a surface mass balance forcing $a(x,t)$, the equations for the ice thickness evolution and SIA ice flow are given by:

$$\frac{\partial H}{\partial t} + \frac{\partial(\bar{u}H)}{\partial x} = a, \tag{1}$$

$$\bar{u} = \frac{2A(\rho g)^n}{n+2} H^{n+1} \left|\frac{\partial h}{\partial x}\right|^{n-1} \frac{\partial h}{\partial x}, \tag{2}$$

where $\rho$ is the density of ice, $g$ is gravitational acceleration, $A$ and $n$ are the rate factor and exponent from Glen's flow law, respectively, and $h(x,t) = H + b$ is the glacier surface elevation for a given bed elevation $b(x)$. Note that parameter choices

in this model will have some effect on the ice flow (e.g. a larger value of $A$ will result in smaller glaciers that respond more quickly to climate forcing) but for reasonable values do not significantly change any of the results in this study. The boundary conditions for the ice thickness $H$ are handled by specifying a Dirichlet or Neumann boundary condition at $x = 0$ and requiring that $H$ goes to zero at the glacier terminus, where the ice front position is a free boundary.

### 2.1.2 Debris dynamics

We assume that debris is homogeneously distributed within the ice with a spatially constant concentration $c$. The debris melts out when the ice melts at the surface and remains on the surface, where it is passively advected with the surface ice flow velocity $u_s = \frac{n+2}{n+1}\bar{u}$, until it reaches the terminus. The evolution of surface debris thickness $D(x,t)$ is represented by:

$$\frac{\partial D}{\partial t} + \frac{\partial (u_s D)}{\partial x} = \phi,$$
(3)

where $\phi$ is the debris source term at the surface given by

$$\phi(a, H) = \begin{cases} 0, & \text{if } a \geq 0 \\ -ca, & \text{if } a < 0 \end{cases}.$$
(4)

Note that for simplicity, we do not account for debris volume changes during melt due to density differences and debris porosity, hence our formulation is different by a constant factor compared to Naito et al. (2000) and Anderson and Anderson (2016). Further, the assumption of uniform debris concentration within the ice means that debris will be present over the entire ablation area and hence our model is representative of extensively debris covered glaciers with debris deposition in the accumulation area close to the ELA or even extending beyond, into the ablation area (e.g. Himalaya).

### 2.1.3 Surface mass balance

We assume that debris-free ice has an elevation dependent surface mass balance $\tilde{a}$ given by

$$\tilde{a}(z) = \min\{\gamma(H + b - \text{ELA}), a_{max}\},$$
(5)

where $\gamma$ is the mass balance gradient, ELA is the equilibrium line altitude, and $a_{max}$ is a maximum mass balance, which limits the accumulation to physically realistic values at very high elevations. A surface layer of debris enhances ice ablation when its thickness $D$ is below a threshold $D_0$. We neglect the effect of enhanced ablation when $D < D_0$ and represent the inverse relationship of surface mass balance with debris thickness (Anderson and Anderson, 2016) as

$$a = \tilde{a}\frac{D_0}{D_0 + D}.$$
(6)

The parameter $D_0$ is chosen based on an Østrem curve that is representative for data from Zmuttgletscher, a medium-sized Alpine glacier (e.g. Nicholson and Benn, 2006; Mölg et al., 2019).

### 2.1.4 Debris boundary condition at glacier front

The choice of boundary condition at the terminus is of critical importance, since the rate at which the supraglacial debris covering the ablation zone leaves the glacier significantly affects the glacier extent (Anderson and Anderson, 2016) and if this is handled incorrectly, it can lead to runaway glacier growth (Konrad and Humphrey, 2000). To ensure that the boundary condition makes sense, it should be consistent with observations and also grid size independent, since the laws of physics should not depend on the choice of discretization.

With these requirements in mind, we set the boundary condition such that the debris leaves the system via a terminal ice cliff, as typically observed at termini of debris covered glaciers (Ogilvie, 1904, see Fig. A1 in the appendix). This can be achieved most easily through an adjustment of the surface mass balance, which we adjust at the point where the ice reaches a critical thickness $H^*$ (Fig. A2). All debris melted out or transported past this point is assumed to slide off of the glacier relatively quickly and is therefore removed from the surface there. This implies that the glacier will always have a small debris-free cliff area at the terminus with clean ice melt. Therefore near the terminus, the surface mass balance $a$ is given by

$$a = \begin{cases} \tilde{a}\frac{D_0}{D_0+D}, & \text{for } x < x^* \\ \tilde{a}, & \text{for } x \geq x^* \end{cases}, \tag{7}$$

where as above, $\tilde{a}$ is the debris-free surface mass balance, and $x^*$ is the location at which the ice thickness $H = H^*$ (with larger $x$ values corresponding to positions further downglacier). In addition, we accounted for fact that the near-terminus ice velocity in SIA goes to zero at a faster rate than is physically realistic by adjusting the velocity here. The mean velocity from the region upglacier averaged over ten ice thicknesses (here about 300 m) is used when computing the debris transport. For more details of the implementation of the terminus parametrization see Appendix A.

We note that our boundary condition is similar to the one implemented in Anderson and Anderson (2016), but our approach differs in that we remove debris beyond a critical thickness (position of ice cliff), whereas in Anderson and Anderson (2016) it is removed from a terminal wedge. Although our ice cliff position is by construction not really grid size dependent, the modelled terminus position shows some dependency on grid size. However, sensitivity tests demonstrate that there is fast convergence with decreasing grid size and the dependency for the 25 m grid size resolution used here essentially vanishes (see Appendix A, Table A1 and supplementary figure A2).

### 2.1.5 Terminus cryokarst features

For some experiments, we attempt to include the effects of melt enhancement from cryokarst features. Observations indicate that ice cliffs and supraglacial ponds commonly occur near the termini of stagnating debris-covered glaciers (Pellicciotti et al., 2015; Brun et al., 2016; Kraaijenbrink et al., 2016; Watson et al., 2017) and are associated with regions that have low driving

stresses (Benn et al., 2012). The driving stress $\tau_d$, representing the weight of the ice column, is given by

$$\tau_d = \rho g H \frac{\partial h}{\partial x}. \tag{8}$$

Using such a dynamic coupling as a first approximation, we couple the initiation of cryokarst features to driving stresses below

a threshold value. Specifically, we define two driving stress thresholds, a maximum $\tau_d^+$ and a minimum $\tau_d^-$ and we introduce a local cryokarst area fraction $\lambda$ which represents the debris-free area associated with ice cliffs and supraglacial ponds. For a driving stress above $\tau_d^+$, the local cryokarst area fraction $\lambda$ is set to zero, which corresponds to no ice cliffs and no supraglacial lakes. For a driving stress below $\tau_d^-$, the local cryokarst area fraction equals a maximum value $\lambda_m$. For driving stress values in between the thresholds, we assume the local cryokarst contribution is linear in $\lambda$, given by

$$\lambda = \begin{cases} 0, & \text{if } \tau_d^+ \leq \tau_d \\ \lambda_m(\tau_d^+ - \tau_d)/(\tau_d^+ - \tau_d^-), & \text{if } \tau_d^- < \tau_d < \tau_d^+ \\ \lambda_m, & \text{if } \tau_d \leq \tau_d^- \end{cases} \tag{9}$$

This dependence of cryokarts area fraction on driving stress is illustrated in Fig. 1 for $\tau_d^- = 60$ kPa, $\tau_d^+ = 110$ kPa, and $\lambda_m = 0.1$.

For the fraction of area where cryokarst is present, we assume that there is no longer an insulating effect on the surface mass balance. Adjusting the local surface mass balance $a$ to account for this gives

$$a = \lambda \tilde{a} + (1 - \lambda) \tilde{a} \frac{D_0}{D_0 + D}. \tag{10}$$

The threshold values $\tau_d^+$ and $\tau_d^-$ are based on the values of the driving stress during advance and retreat in the cryokarst–free case and are chosen such that $\tau_d$ only drops below the upper threshold $\tau_d^+$ once the glacier begins to stagnate during retreat. From numerical simulations of retreat, we determined that realistic values for the thresholds are $\tau_d^+$ between 100 and 125 kPa and $\tau_d^-$ between 50 and 75 kPa. For more details, see Appendix B.

### 2.1.6    Model setup numerical implementation

The coupled dynamic system described above is solved using standard finite differences and discretizations for the ice flow, similar to that first described by Mahaffy (1976), coupled with centred differences for the debris transport. Care is taken to ensure that each time step fulfills the Courant-Friedrichs-Levy (CFL) condition, which is necessary for the numerical stability of the method (Courant et al., 1928). Essentially, the CFL condition limits the length of each time step so that information from

any computational cell can propagate only as far as its nearest neighbours. Importantly, the boundary condition at the glacier terminus requires interpolation to determine the exact location of the critical ice thickness $H^*$ and to weight the surface mass balance forcing accordingly in the corresponding grid cell.

In the results that follow, all computations are performed using a bed consisting of a headwall with a slope of 45° followed by

a linear bed with slope of roughly 6°. All model constants are shown below in Table 2.

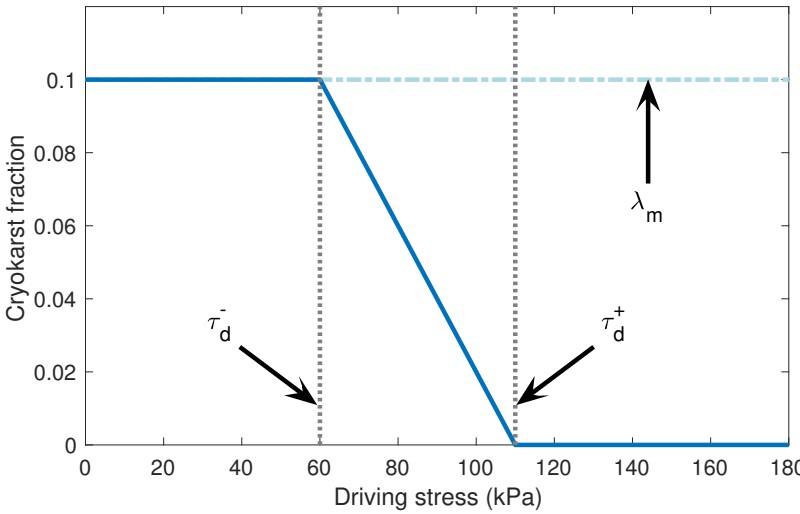

**Figure 1.** Cryokarst area fraction $\lambda$ for a range of driving stress values $\tau_d$ for the case of $\tau_d^- = 60$ kPa, $\tau_d^+ = 110$ kPa, and $\lambda_m = 0.1$.

## 3 Modelling results

Our goal is to better understand how the transient response of a debris-covered glacier is different than that of a debris-free glacier and what effect cryokarst features have on this transient response. To do this, we perform a series of numerical experiments consisting of applying step changes or random climate histories in the climate forcing for glaciers with variable debris concentration and hence various levels of surface debris and analyze the resulting volume and length response.

First we examine the steady state case for two climates and four different debris concentrations in Section 3.1. Then in Section 3.2, we examine the transient behaviour of glaciers that move from one steady state to another after a step change in climate for the case of debris concentration $c = 0.25\%$. Next, we simulate a random climate forcing for a duration of 5000 years for glaciers with different debris concentrations and examine the resulting transient volume and length response, found in Section 3.3. We next examine the effect of introducing cryokarst features near the terminus on the transient response to a step change in climate in Section 3.4. Finally, we have a second look at the transient response to random climate forcing when cryokarst is present in Section 3.5. An overview of these experiments is found in Table 1, with reference to the relevant figures in the text. Table 2 summarizes the parameter values used in the numerical model.

### 3.1 Steady state glacier extent

As a baseline for understanding the glacier's response to a changing climate, we first examine steady state features for the two climate extremes of our study, ELA = 3000 m and 3100 m, which are representative of the climate for a medium-sized Alpine

**Table 1.** Summary of modelling experiments performed.

| No. | Description | Section | Figures |
|---|---|---|---|
| 0 | Baseline: steady-states at ELA = 3000, 3100 m | 3.1 | 2 |
| 1 | Transient response due to step change between steady-states | 3.2 | 3,4,5 |
| 2 | Random climate forcing | 3.3 | 6 |
| 3 | Transient response with cryokarst | 3.4 | 7,8 |
| 4 | Random climate forcing with cryokarst | 3.5 | 9 |

**Table 2.** Values used for the model parameters.

| Parameter | Name | Value | Units |
|---|---|---|---|
| ELA | Equilibrium line altitude | $3000 - 3100$ | m |
| $\rho$ | Density of ice | 910 | $kg\,m^{-3}$ |
| $g$ | Gravitational acceleration | 9.80 | $m\,s^{-2}$ |
| $c$ | Debris volume concentration | $0 - 0.005$ | |
| $A$ | Flow law parameter | $1 \times 10^{-24}$ | $Pa^{-3}\,s^{-1}$ |
| $n$ | Glen's constant | 3 | |
| $D_0$ | Characteristic debris thickness | 0.05 | m |
| $a_{max}$ | Maximum surface mass balance | 2 | $m\,yr^{-1}$ |
| $\gamma$ | Surface mass balance gradient | 0.007 | $yr^{-1}$ |
| $H^*$ | Terminal ice thickness threshold | 30 | m |
| $\lambda_m$ | Maximum cryokarst fraction | $0 - 0.2$ | |
| $dt$ | Time step | 0.01 | yr |
| $dx$ | Spatial discretization | 25 | m |
| $\tau_d^+$ | Upper cryokarst driving stress threshold | $100-125$ | kPa |
| $\tau_d^-$ | Lower cryokarst driving stress threshold | $50-75$ | kPa |
| $\theta$ | Bed slope | 0.1 | $m\,m^{-1}$ |
| $\theta_c$ | Headwall slope | 1 | $m\,m^{-1}$ |

glacier during the last century. Although in reality, glaciers never attain a true steady state due to a constantly changing climate, equilibrium conditions are useful for theoretical studies because they provide well-understood rest states around which glacier fluctuations can be more easily studied. As for debris-free glaciers, a steady state is defined as the point at which, for a fixed climate, the glacier geometry no longer changes with time. Additionally, in our modelling there is a further requirement that the debris flux entering the glacier also must leave the glacier surface at the terminus.

Figure 2 shows the steady state glacier surface and bed profiles, velocity profiles, and debris thickness profiles for the debris-

free case as well as for the debris concentrations of $c = 0.1, 0.25$, and $0.5\%$. The glacier profiles in Fig. 2a and 2d show the expected behaviour of higher debris concentration leading to longer, larger glaciers. A debris concentration of only $0.1\%$ almost doubles the glacier length compared to the clean ice case. Note that the glacier geometry in the debris-free part above the ELA is almost identical for all cases (with a small difference when greater debris cover results in a more elongated glacier with a lower surface slope at the ELA) and hence independent of debris concentration. The surface mass balance for the

debris-covered glaciers no longer decreases linearly with elevation but is instead controlled primarily by the debris thickness and strongly reduced over most of the ablation area, as shown in Fig. 2b and 2e. Surface velocities generally decrease with increasing debris thickness along the glacier, as seen in Fig. 2c and 2f.

An interesting observation here is that for a fixed climate, the debris thickness profile in steady state appears to be approx-

imately independent of concentration, while the glacier extents differ strongly. This is discussed further in Sec. 4.5.

### 3.2   Transient response between steady states

Next we analyze the response to a step change in the climate forcing. Figure 3 shows the transient volume and length changes due to ELA step changes of $\pm 100$ m. In Fig. 3b, glacier volume time series show the response time dependence on debris concentration, with the expected result that higher debris concentration leads to longer volume response time. Here, filled-in

squares denote the $e$-folding volume response time (Jóhannesson et al., 1989; Oerlemans, 2001), which is the time it takes to reach $1 - 1/e \simeq 63\%$ of the total volume change. The values of the numerical volume response time are shown in Table 3 in the columns marked $T_{num}$. In general, the volume response times are strongly increased for debris-covered glaciers compared to the debris-free case. A more detailed discussion of volume response time follows in Section 4.3.

In Fig. 3c, the length times series allow for a comparison with the length response time. For the case of glacier advance, shown on the right side of the plot starting at $T = 1000$ years, the form of the length change is similar to that of the volume change: a slow but steady increase leading asymptotically to a steady state. However, the retreat phase, shown for $T = 0$ to 1000 yr, is contrasting this response behaviour. Here, we see a clear lag in length response, which gets stronger for larger debris concentrations. The lag is so pronounced that when the glaciers have reached their respective $e$-folding volumes, denoted as

filled-in squares, they are still approximately at their pre-step change extent.

To show the difference in length versus volume response more clearly, we closely examine one debris-covered glacier, with $c = 0.25\%$ debris concentration, and contrast its response with the debris-free case. In Figure 4, the normalized volume and length are plotted together for each glacier, where we have set the cold (ELA = 3000 m) steady state volume $V = 1$ (length

$L = 1$) and warm (ELA = 3100 m) steady state volume $V = 0$ (warm length $L = 0$) for ease of comparison. For the debris-free case, shown in Fig. 4a, the volume and length curves follow each other closely but there is a small but noticeable time lag

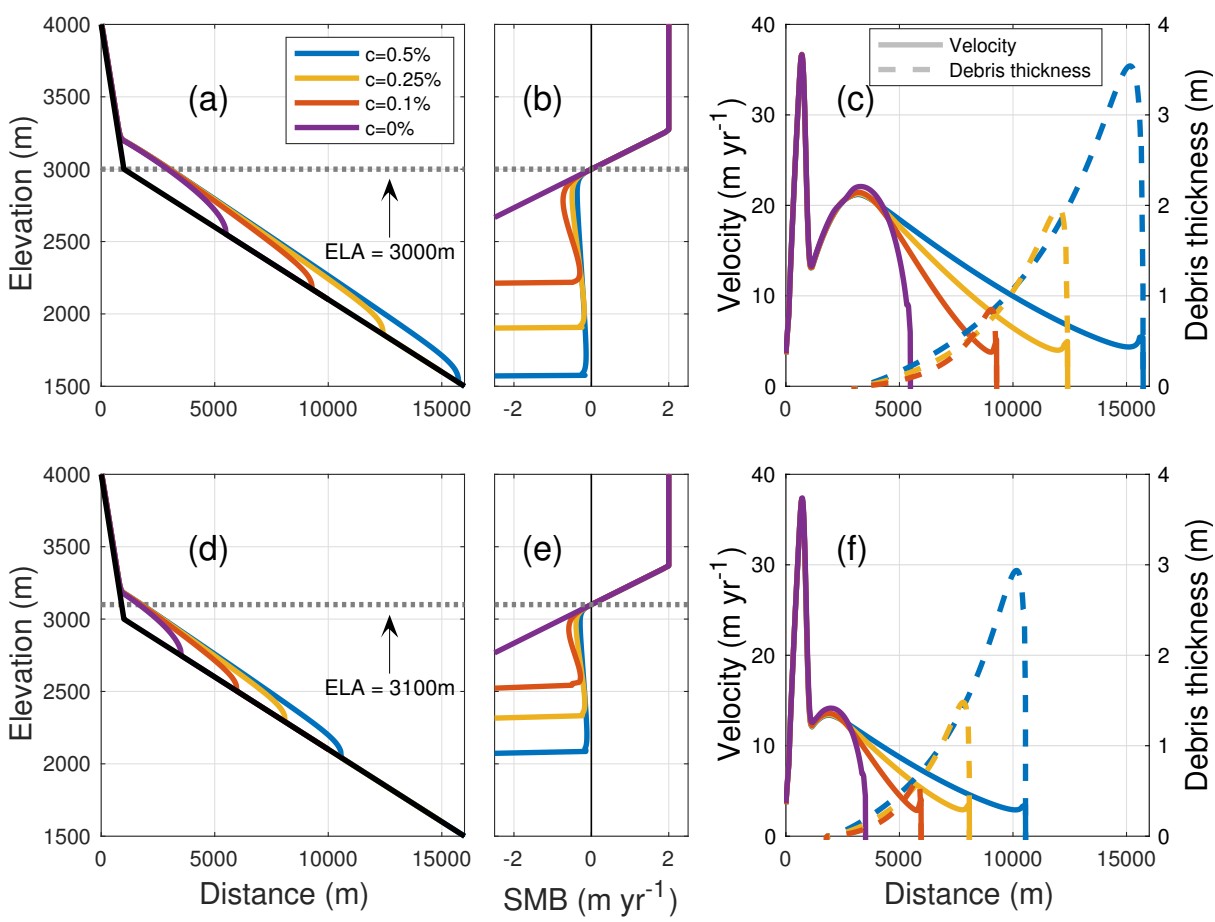

**Figure 2.** Steady state glacier geometry profiles (a and c), and profiles of surface velocity (solid lines) and debris thickness (dashed lines) (b and d), corresponding to ELA = 3000 m and 3100 m for four different debris concentrations.

between the volume response and the length response, which is more evident during retreat. The debris-covered response, in Fig. 4b, shows a substantial lag time between the volume response and the length response. This lag in the length response is, at roughly 250 years, much larger during the retreat but is also observable during the advance, where a 50 year lag is observed at the onset of advance. An additional difference between the glaciers is found near the end of the retreat phase. The transient debris-covered glacier volume overshoots the final steady state volume, observable starting just before $T = 500$ yr in Fig. 4b, before recovering to its final volume. During this overshoot and recovery in volume, the transient glacier length monotonically decreases and never goes below its final steady state length. In contrast, the transient debris-free glacier volume has no over-shoots: it monotonically decreases during retreat.

In Figure 5, we again compare the debris-free case with the $c = 0.25\%$ debris-covered case but this time we look at the

respective glacier thickness profiles during retreat. To facilitate comparison across spatial and temporal scales, we plot the normalized glacier thickness profiles for equivalent relative transient evolution times during retreat for both glaciers. In Fig. 5a, we see that immediately as the debris-free glacier thins, it also retreats in a roughly uniform way with thinning approximately

matched by reduction in glacier extent. This can be thought of as a manifestation of a volume-area scaling law $V = cA^\gamma$ (e.g. Bahr et al., 1997), which essentially says that a debris-free glacier volume is linked to its area by a power law. Note that for our flowline model, the equivalent scaling law is a relationship between area and length. The debris-covered glacier profile shown in Fig. 5b does not follow the same pattern. As the glacier thins during the period of relative time $\Delta t = 0$ to $\Delta t = 0.6$, there is no discernible change in the glacier extent. Initially, most of the thinning occurs in the upper half of the ablation zone but

ceases there rapidly after $\Delta t = 0.1$ to $0.2$ relative time. By $\Delta t = 0.6$, the entire region from relative length $x = 0$ to $x = 0.6$ is at or slightly below its final steady state ice thickness even though the original glacier extent has not changed yet. Only by $\Delta t = 0.8$ do we finally see the glacier terminus start to retreat, with the last roughly 35% of the glacier appearing as a thin, not very dynamic, and soon to be disconnected terminus (see Fig. S1 in the supplementary material for corresponding velocity profiles). Although the velocity is nonzero, this type of terminus is often described as stagnating because dynamic replacement

of ice is close to zero and hence the local thinning rate is roughly equal to the local surface mass balance. The loss of this stagnant terminus results in a large, rapid decrease in ice volume and a corresponding rapid retreat (e.g. the blue curves in Fig. 3b and 3c at around t = 600 yr). In the final 20% of the total retreat time, the stagnant terminus has completely disappeared and the central region that had previously over-thinned has now recovered to its steady state thickness and has become the new terminus. Note that it is this rapid loss of We revisit this interesting terminus behaviour in the section 4.1 below.

**Table 3.** Comparison of numerical $e$-folding volume response times (in years) due to step changes between steady states at ELA = 3000 m and ELA = 3100 m for different debris concentrations. The columns marked $T_{num}$ represent the numerical volume response time and the other columns represent the theoretical estimate of Jóhannesson et al. (1989) developed for clean ice using different methods of calculating the surface mass balance at the terminus (see section 4.4 for details).

| | $\tau_v$ – retreat | | | | | $\tau_v$ – advance | | | | |
|---|---|---|---|---|---|---|---|---|---|---|
| $c$ | $T_{num}$ | $T_1$ | $T_2$ | $T_3$ | $T_4$ | $T_{num}$ | $T_1$ | $T_2$ | $T_3$ | $T_4$ |
| 0 | 77 | 52 | – | – | – | 133 | 108 | – | – | – |
| 0.1 | 154 | 29 | 456 | 54 | 222 | 265 | 48 | 694 | 87 | 340 |
| 0.25 | 256 | 22 | 646 | 42 | 207 | 396 | 34 | 955 | 66 | 321 |
| 0.5 | 385 | 17 | 800 | 34 | 179 | 529 | 25 | 1237 | 50 | 273 |

## 3.3   Response to random climate forcing

We have investigated debris-covered glacier response to step changes in the climate and it is natural to query whether these results will have any bearing on a more realistic fluctuating climate input. To investigate this issue in a somewhat less idealized setting, we initialize the model to a steady state corresponding to an ELA of 3050 m. Then we force the model using a varying

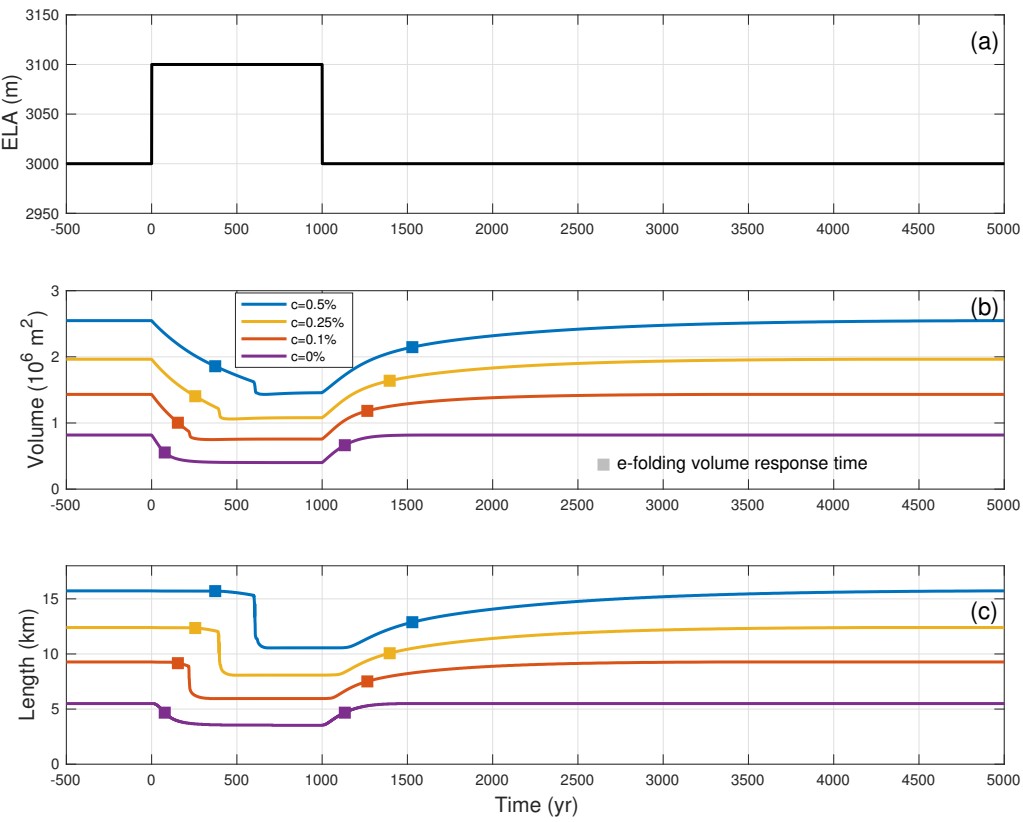

**Figure 3.** Step change in ELA between steady states (a) leading to transient volume response (b) and transient length response (c) for glaciers with different debris concentrations (coloured lines). The filled-in squares in both (b) and (c) represent the $e$-folding volume response time.

climate signal consisting of a 5000 year long time series made up of random fluctuations between ELA = 3000 m and ELA = 3100 m which corresponds in the Alps to a change in air temperature of about $0.8°C$ (Linsbauer et al., 2013). The fluctuations occur at fixed intervals of 100 years, which is close to but a bit larger than the clean ice response time during retreat, and they have a mean of ELA = 3050 m. This random climate forcing is shown in Fig. 6a and the respective transient volume and length time series are shown in Fig. 6b and 6c for a debris-free glacier and three debris-covered glaciers with different debris concentrations.

The behaviour of the debris-free glacier (purple line at the bottom of Fig. 6b and 6c) exhibits a relatively rapid volume and length response, which can be seen by how quickly the solid curve, representing the transient, moves back towards the dashed line, which represents the steady state value for the mean climate of ELA = 3050 m. For the debris-covered glaciers with lower

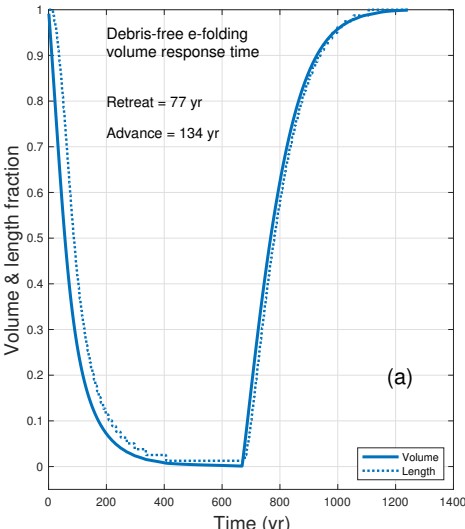 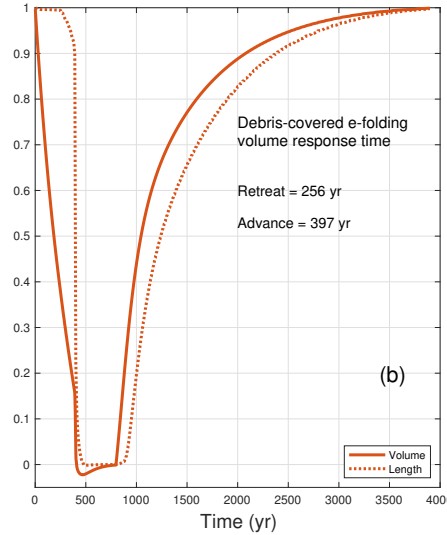

**Figure 4.** Normalized transient volume and length response for a step change in ELA between two steady states for (a) a debris-free glacier and (b) a debris-covered glacier with $c = 0.25\%$.

concentration (red and yellow lines in Fig. 6b), the volume responds only marginally more slowly. The difference is more
pronounced for the glacier with the greatest debris concentration (blue line) where the transient volume never goes below the mean climate steady state volume beginning from $T = 2800$ yr. Note that the light blue shading corresponds to colder than average time periods and white shading corresponds to warmer than average time periods.

The asymmetric response is much more pronounced in the transient length time series. While the glacier with the lowest
debris concentration (red line in Fig. 6c), exhibits a marginally slower response compared to the debris-free case (solid purple line), the glaciers more heavily laden with debris, shown in yellow and blue, have transients lengths that are almost throughout more extended than the mean climate steady state length and tend to advance more than retreat. This is especially true for the $c = 0.5\%$ case, which after 5000 yr is more than 1 km longer than one would expect for the mean value climate. Hence, due to the lag in the length response to a warming climate, debris-covered glaciers preferentially show the effects of the colder
climate. Put another way, debris-covered glaciers remember periods of cold climate more than warm ones. This also suggests that time-averaged length of a debris-covered glacier under random climate forcing will be longer than the steady state length for the equivalent constant climate forcing.

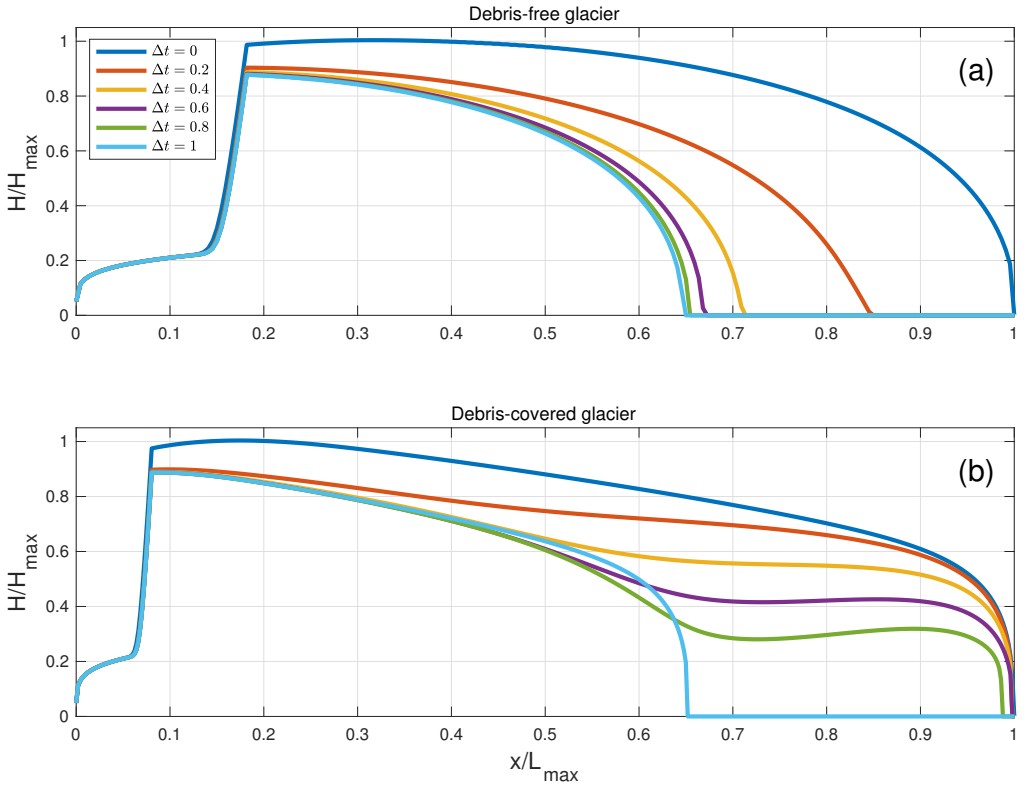

**Figure 5.** Glacier thickness profiles relative to the maximal initial thickness for (a) a debris-free glacier and (b) a debris-covered glacier with $c = 0.25\%$ at different times during a transient retreat. The different coloured lines refer to the time relative to the time it takes to retreat to steady state, where a time of $t = 0.1$ corresponds in the debris-free case to about 38 years and in the debris-covered case to about 40 years.

### 3.4 Effect of cryokarst on response

Most of the debris-covered glaciers observed in the present day have varying amounts of ice cliffs and supraglacial ponds present on their tongues which are known to enhance surface ablation (Benn et al., 2012). However, the long-term effect of such cryokarst features on thinning and glacier dynamics is poorly understood. With this in mind, we repeat the above experiments for four debris-covered glaciers, all with a medium debris concentration $c = 0.25\%$ by including the dynamic cryokarst model introduced in Section 2.1.5 and perform runs for different maximum local cryokarst area fraction of $\lambda_m = 0, 5, 10$, and $20\%$ (consistent with observations from Mölg et al., 2019; Steiner et al., 2019; Anderson et al., 2021). Driving stress thresholds of $\tau_d^+ = 110$ kPa and $\tau_d^- = 60$ kPa are used here. Since we dynamically couple the onset and intensity of melt enhancement from cryokarst to the driving stress using equations (9) and (10), the effect of cryokarst is only felt during periods of mass loss and we focus exclusively on this in Figure 7. The purple lines in Fig. 7a and 7b correspond to the case with no cryokarst

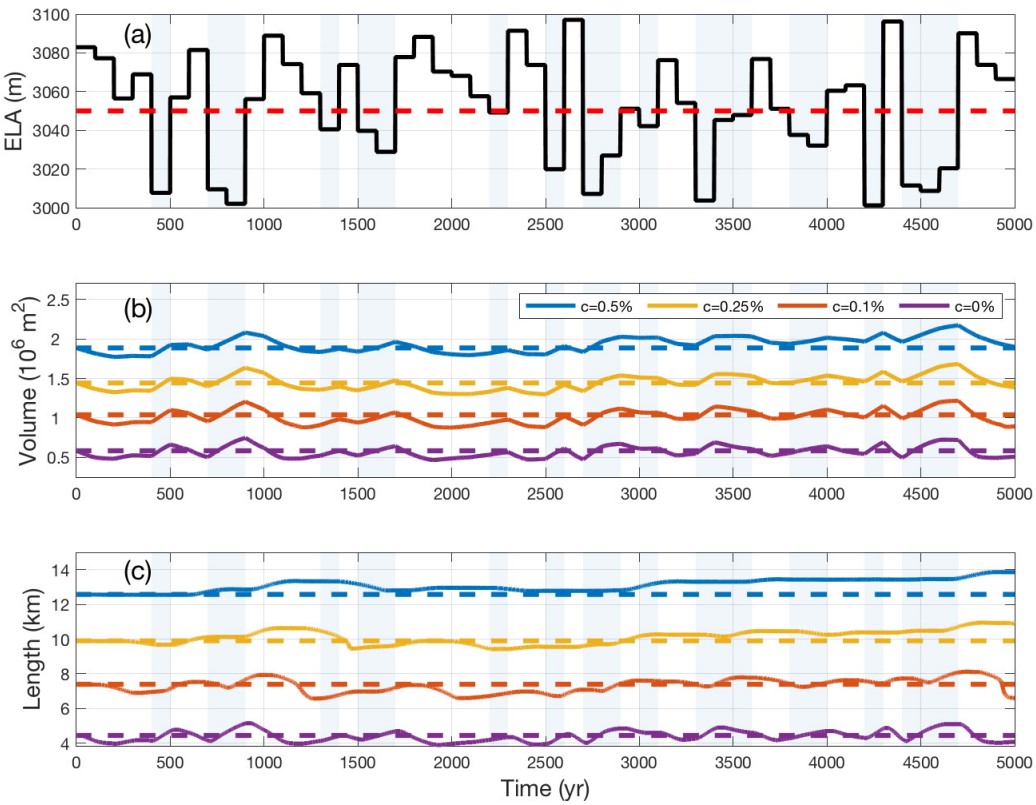

**Figure 6.** Random climate forcing (a) and the corresponding transient volume response (b) and transient length response (c) for glaciers of different debris concentrations. The dashed lines represent (a) the mean value climate of ELA = 3050 m and the corresponding steady state (b) volume and (c) length. The light blue background shading represents temporal periods during which the climate forcing is colder than the mean climate.

features and therefore they show the same retreat as the yellow lines in Fig. 3b and 3c.

The addition of cryokarst has a noticeable effect on both the volume and length response of a debris-covered glacier. In Fig. 7a, there is a clear reduction in $e$-folding volume response time of a couple of decades visible (see Table 4 for the exact values), which is more pronounced with the presence of enhanced cryokarst. The effect on glacier length response is even more striking, with a difference of more than a century between the timings of the onset of retreat. The actual amount of equivalent bare ice for each glacier is shown as a percentage of the entire ablation zone area in Fig. 7c. Even for the smallest amount of

cryokarst modelled, which accounts for only 2% of equivalent bare ice in the ablation area, there is a shortening of roughly 70 yr in the timing of the main phase of retreat. Despite this evident effect the presence of cryokarst has on length response,

there is still a significant lag observed compared to the clean ice case, as in all modelled cases the glaciers are still at their maximum pre-step change extents even by the respective $e$-folding volume response times. Note that the choice of driving stress thresholds affects the strength of the cryokarst effect on the response (see Fig. S2 and S3 in the supplementary material).


To aid in the visualization of the effect of cryokarst on the transient glacier dynamics, we plot driving stress, cryokarst area fraction, and melt rate in Fig. 8 at five different time steps, spaced fifty years apart, corresponding to the case of $\lambda_m = 10\%$. As the driving stress drops below the upper threshold $\tau_d^+$, shown in Fig 8a, the cryokarst area fraction is observed to increase, shown in Fig 8b. This leads to an increase in melt rate, depicted in Fig 8c. The maximum melt rates occur at $t = 250$ and

300 yr, shown in yellow and purple, when the driving stress is at or below the lower threshold $\tau_d^-$ and the corresponding cryokarst area fraction $\lambda$ at the terminus achieves the maximum value $\lambda_m$. This increased melt rate contributes to a rapid retreat, corresponding to the green line for $t = 350$.

**Table 4.** Comparison of numerical $e$-folding volume response times (in years) due to step changes between ELA = 3000 m and ELA = 3100 m for a debris concentration of $c = 0.25\%$ and with the presence of several different values of maximum terminal cryokarst fraction $\lambda_m$.

| Debris conc. $c$ | Max. cryo. fraction $\lambda_m$ | Vol. resp. time $\tau_v$ |
|---|---|---|
| 0.25 | 0 | 256 |
| 0.25 | 0.05 | 234 |
| 0.25 | 0.1 | 219 |
| 0.25 | 0.2 | 200 |

### 3.5    Response to random climate forcing with presence of cryokarst

Figure 9 shows the same random climate forcing experiment with $c = 0.25\%$ as in Fig. 6 but cyrokarst with driving stress

thresholds of $\tau_d^+ = 110$ kPa and $\tau_d^- = 60$ kPa, and four different maximum local cryokarst area fractions $\lambda_m = 0, 5, 10$, and 20%. As before, the $\lambda_m = 0$ glacier, corresponding to the purple curves in Fig. 9, is identical to the results already plotted in Fig. 6. in yellow. Since there is no dynamical effect during periods of advance, we do not see much difference in either the volume or length change rate in colder climate regimes; see in particular between 3000 and 5000 yr model time. However, during retreat the difference is visible especially during the warm climate dominated period of $T = 1300$ to 3000 yr. In Fig. 9b

and even more clearly in Fig. 9c, the transient volume and length of the cryokarst covered glaciers exhibit a shorter memory and are therefore able to retreat much more quickly than the corresponding cryokarst-free glacier. Despite this faster response, all of the modelled debris-covered glaciers still respond with much more delay than a debris-free glacier with the same climate forcing, as shown in the black dotted line in Fig. 9b and 9c, which has been rescaled in the magnitude for both length and volume for ease of comparison. As noted above, the timing in onset of retreat is rather sensitive to the choice of the upper and

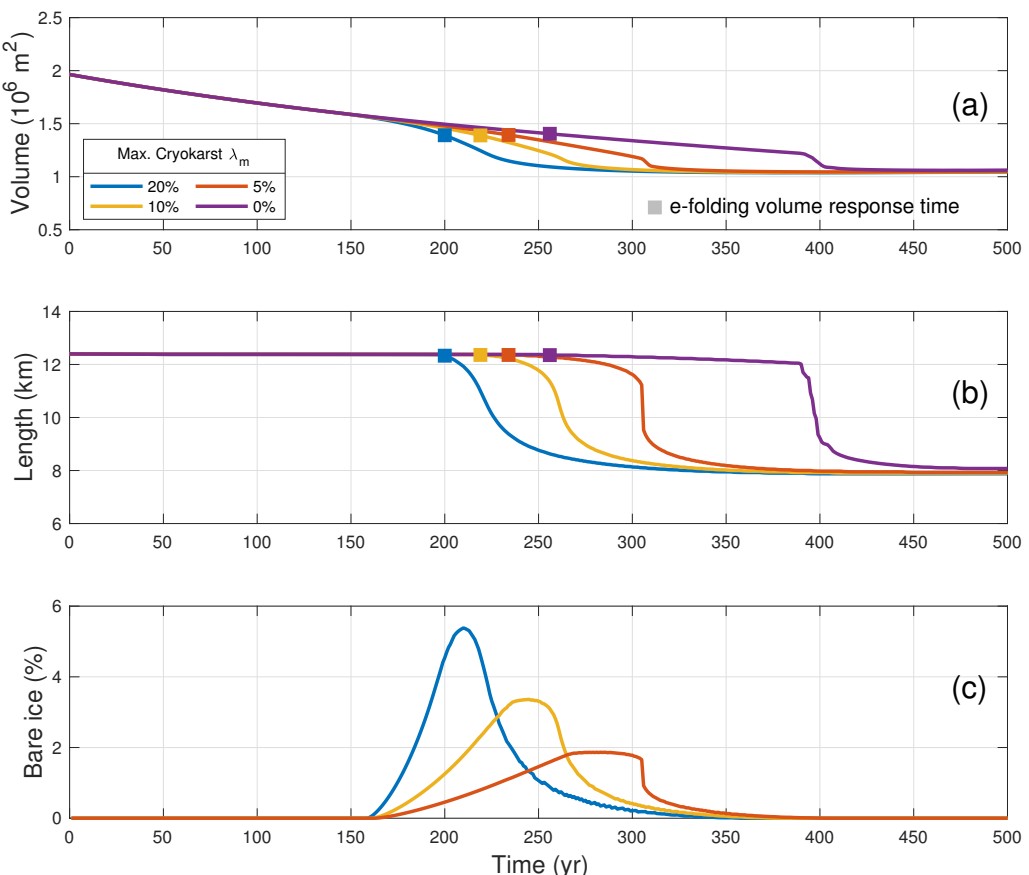

**Figure 7.** Transient volume response (a) and length response (b) for debris-covered glaciers with terminal cryokarst features retreating from steady state after a 100 m step change in ELA. Each colour represents a different value of the maximum cryokarst percentage $\lambda_m$. In all cases, the debris concentration is $c = 0.25\%$ and the driving stress thresholds are $\tau_d^+ = 110$ kPa and $\tau_d^- = 60$ kPa. The filled in squares in (a) and (b) represent the $e$-folding volume response time. The percentage of total debris-covered length that has a bare ice equivalent surface mass balance due to the presence of cryokarst is shown in (c).

lower driving stress thresholds. To compare with results using different threshold values, please refer to supplementary figures S4 and S5.

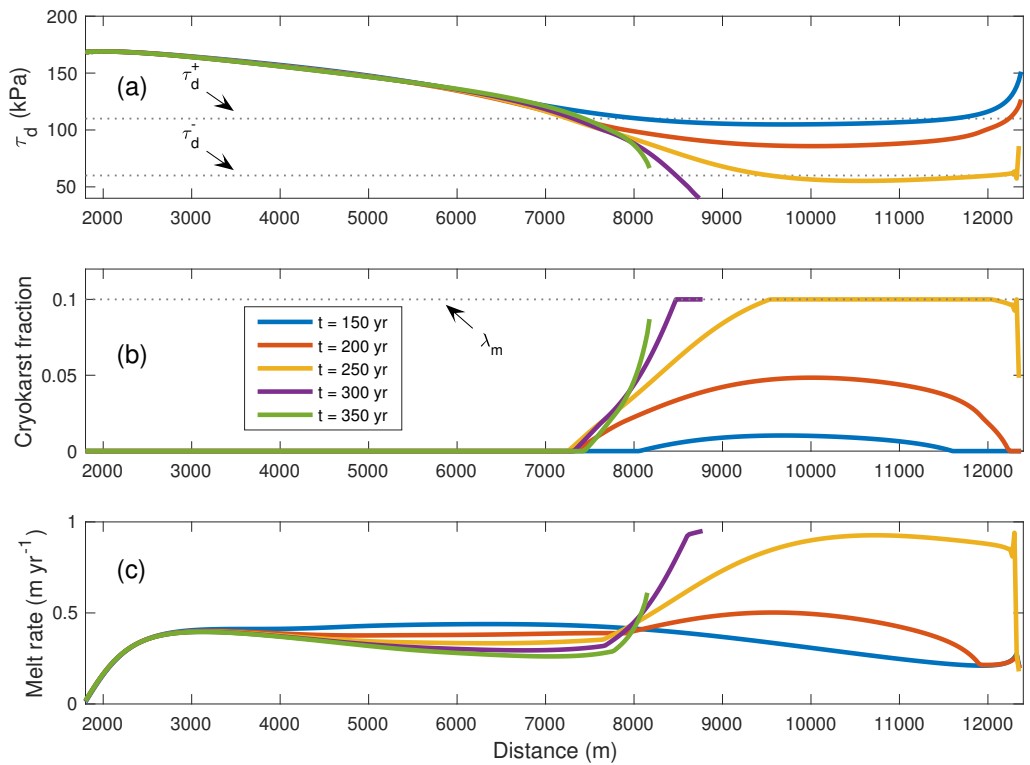

**Figure 8.** Profiles of driving stress (a), cryokarst fraction (b), and melt rate (c) at five different time steps for the case of maximum cryokarst area fraction $\lambda_m = 10\%$. The dashed lines in (a) denote the driving stress thresholds used in the cryokarst parameterization.

## 4 Discussion

We explored the transient response of a debris-covered glacier to changes in climate forcing using a flowline model that couples ice flow with debris melt out and advection and also includes an ad-hoc representation of the effects of dynamically coupled
cryokarst features at the glacier terminus. Several interesting results related to dynamics were obtained, which we discuss separately in light of observational data, previous studies, and model limitations.

### 4.1 Terminus behaviour during transient response

The results of our numerical experiments indicate that debris-covered glaciers have an asymmetric response to climate forcing, with a visible lag in response during a retreat, and that the magnitude of the lag is reduced in the presence of terminal
cryokarst. To better understand this behaviour, we further examine the debris-covered terminus region during advance and retreat and consider the relative magnitudes of surface mass balance $a$ and flux divergence $\partial Q/\partial x$ on the rate of thickness change

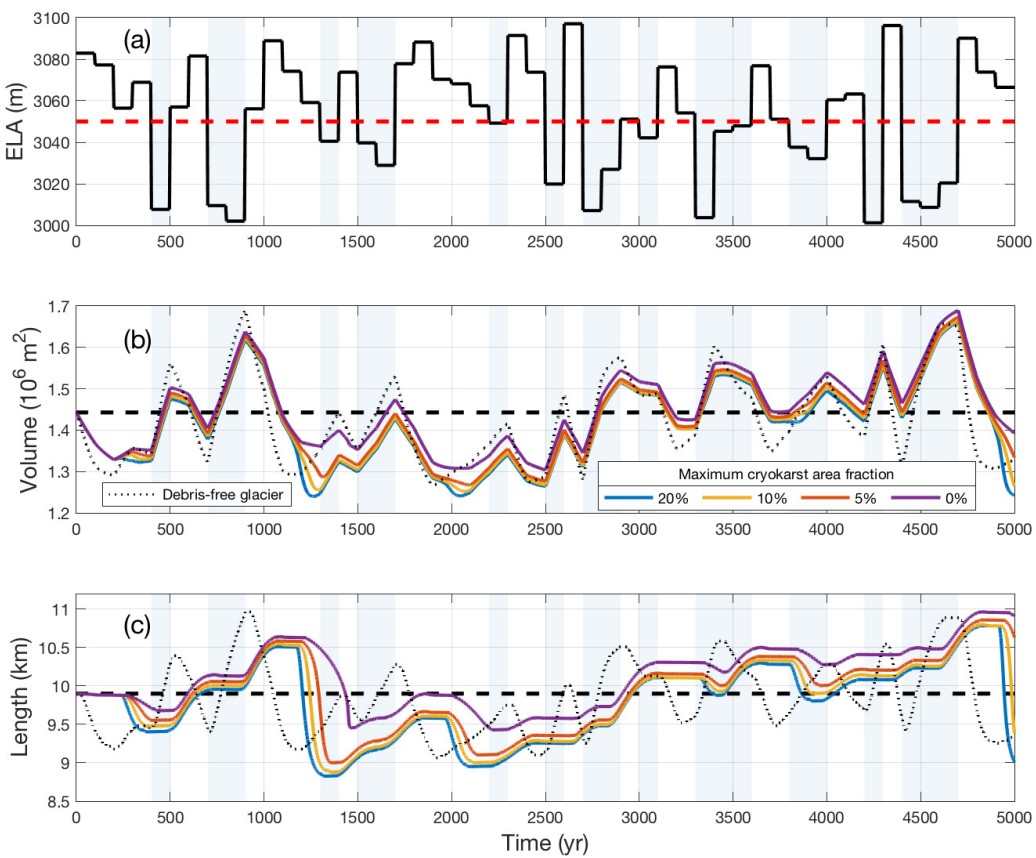

**Figure 9.** Random climate climate forcing (a) and the corresponding transient volume response (b) and transient length response (c) for glaciers of different maximum cryokarst fraction $\lambda_m$. The dashed lines represent (a) the mean value climate of ELA = 3050 m and the corresponding steady state (b) volume and (c) length. The light blue background shading represents temporal periods during which the climate forcing is colder than the mean climate. In all cases, the debris concentration is $c = 0.25\%$ and the cryokarst driving stress thresholds are $\tau_d^+ = 110$ kPa and $\tau_d^- = 60$ kPa.

$\partial H/\partial t$. Anderson et al. (2021) used a similar approach to study thinning at the terminus but here we are primarily interested in the retreat rate. Fig. 10 shows the components of the mass conservation equation (1) for the moving region consisting of the last 200 m of debris-covered area for the cases of advance, retreat, and retreat with maximum local cryokarst area fraction $\lambda_m = 5\%$ and driving stress thresholds $\tau_d^+ = 110$ kPa and $\tau_d^- = 60$ kPa. The initial condition for all three panels is steady state for an ELA = 3050 m, with the advance and retreat due to ELA step changes of $\pm 50$ m at $T = 0$.

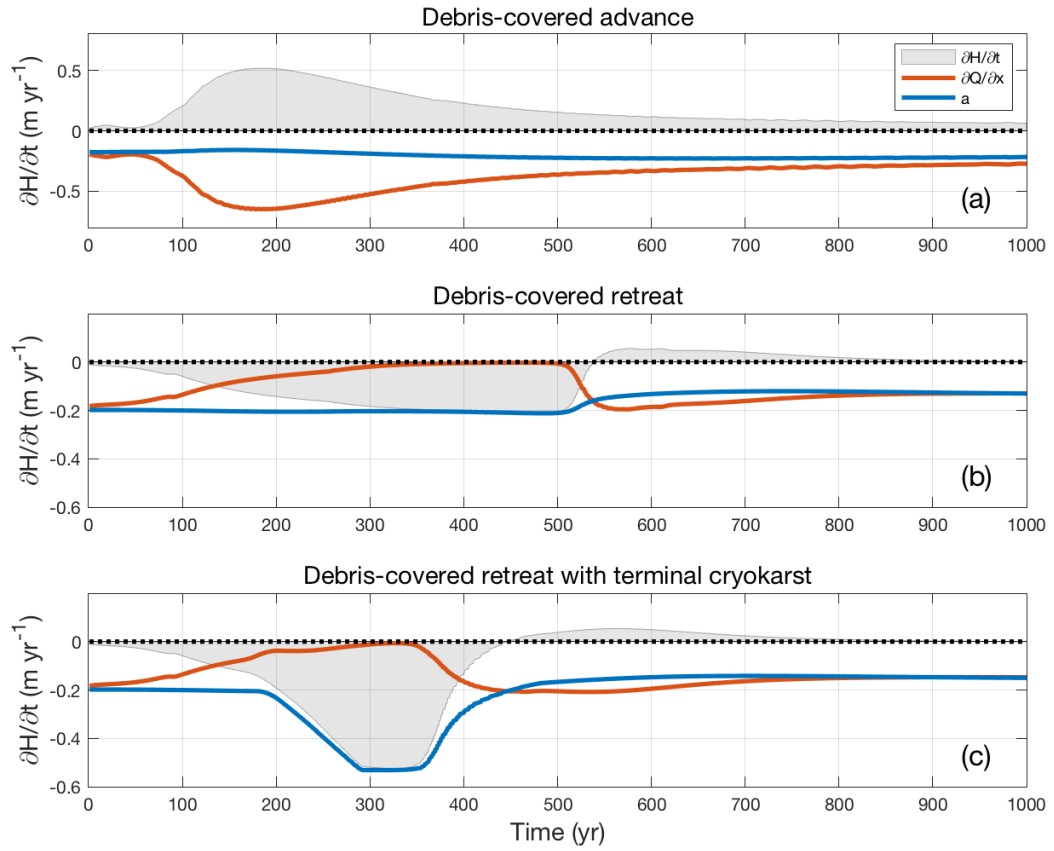

**Figure 10.** Thinning rate, flux divergence, and surface mass balance averaged over the final 200 m of the glacier terminus before the terminus during advance (a), retreat (b), and retreat with cryokarst (c), for $c = 0.25\%$, $\lambda_m = 0.05$, $\tau_d^+ = 110$ kPa, and $\tau_d^- = 60$ kPa. In all cases, the initial condition is a steady state at ELA = 3050 m followed by a 50 m step change in ELA at time $t = 0$.

In each plot, the grey region represents the thickening or thinning rate, with the area under the curve representing the total thickness change of the last 200 m during the entire 1000 yr of the advance or retreat. In all three cases, during the first 200

yr the surface mass balance $a$, plotted in blue, does not change appreciably from the pre-step change value of $a = -0.2$ m yr$^{-1}$. This is because the debris at the terminus is thick enough to make the glacier here relatively insensitive to small changes in climate forcing. For the advancing glacier, depicted in Fig. 10a, the change in surface mass balance is minimal and the thickening rate is driven by an increase in the magnitude of the flux divergence (red line), that peaks at around $T = 150$ yr, the time taken for the increased ice flux to propagate down the glacier to the terminus.


For the retreating glacier, shown in Fig. 10b, the thinning rate is clearly driven by a decrease in flux divergence, which eventu-

ally drops to zero at roughly $T = 300$ yr, at which point the glacier terminus stagnates. It remains so until roughly $T = 500$ yr, when the total amount of thinning, almost equal to the local surface mass balance, is large enough that the stagnant terminus finally disappears. After this, there is a small amount of thickening at the terminus as the glacier readjusts to the overshoot caused by the collapse of the stagnant terminus.

When a small amount of cryokarst features is added to the terminus during retreat, representing at most roughly 2% of the total debris-covered area (Fig 10c), the glacier behaves identically to the cryokarst–free case up until roughly $T = 190$ yr. From then on the terminus dynamics become stagnant enough that the cryokarst features begin to develop and within several decades cause an increase in the melt rate by a factor of more than two. This significantly speeds up the thinning on the tongue and hence the retreat rate, with the bulk of the retreat completed about 100 yr earlier than in the cryokarst–free case.

## 4.2 Debris-covered glacier memory

We showed that the memory of a debris-covered glacier is selective, exhibiting an effective hysteresis, with periods of relatively cold climate having a sustained effect on the volume and in particular on the length. Strictly speaking this is not a true hysteresis since if the glacier is allowed a lengthy relaxation period of several centuries, the resulting equilibrium is independent of history. Some previous numerical simulations of the transient response of debris-covered glaciers focused only on the effects of sudden debris input in the form of an avalanche (Vacco et al., 2010; Menounos et al., 2013). Such a one-time debris input leads to an advance in glacier extent and foreshadows the results of our study, where a constant debris source and changing climate forcing gives rise to a more complex response.

The glacier termini seem to struggle to retreat in warmer periods even if they are sustained over a century, and hence debris covered glaciers have the tendency to either advance or stagnate in a century-scale fluctuating climate (Fig. 6). This also means that for debris covered glaciers, no unique glacier length exists for a given climate, but rather that the length of debris covered glaciers is determined by the history of repeated cold phases. Furthermore, debris-covered glaciers under random climate forcing are expected to have a longer average length than the steady state length corresponding to the equivalent constant climate forcing. These are novel modelling results which have important implications not only for the observed present day extended extents of debris-covered glaciers but also on historical reconstructions. For example, inferences of past climate from historical glacier extents that do not take into account the asymmetric memory of debris-covered glaciers risk misrepresenting the climate as being colder than it actually was (Clark et al., 1994).

Observational data (Quincey et al., 2009; Scherler et al., 2011; Ragettli et al., 2016) show that many debris-covered glaciers have strongly extended and stagnating tongues which is consistent with our modelling and our interpretation that debris-covered glaciers remember rather the colder climates of the past and are therefore quite far out of balance with the present climate. However, since the observational record is not long enough to provide data on meaningful timescales and is heavily biased towards retreating glaciers, it is currently only possible to study this phenomenon fully using numerical experiments.

Note, that this asymmetric response to climate forcing is much more pronounced for the adjustment in glacier length than in volume. In the random climate experiments, volume change and hence average thinning behaviour are surprisingly similar for all debris concentrations and the clean ice case (Fig. 6), which agrees with the general observations of relatively high mass
loss despite the occurrence of substantial debris cover (Pellicciotti et al., 2015; Brun et al., 2018). Such rapid mass loss is governed by two processes. Initially, the warming has a strong impact on the upper accumulation area where debris is still thin. Then the lower tongue with thick debris cover stagnates and dynamic ice replacement diminishes ($\partial Q/\partial x$ goes to zero, Fig. 10b) and hence the ice simply melts away. As this stagnant area is extensive the related total volume loss is therefore substantial.

The results presented used a random climate forcing with a particular ELA range and time interval between random step changes. Different climate forcing signals are possible, as are many different shapes and sizes of glaciers with varying debris thickness profiles. However, the general qualitative results are expected to be the same, though for example a longer time interval between random steps (approaching the debris-covered response time) will reduce the debris-covered memory effect. This is indicated in the random climate experiments by the ability of the terminus to retreat in response to several successive
warm periods (several centuries), as shown in Fig. 6. An obvious extension of this work is to undertake a detailed sensitivity analysis, using a variety of climate signals, glacier geometries, and debris thickness profiles, in order to better understand the conditions under which this selective memory effect becomes significant.

### 4.3   Cryokarst effect in modulating response

Our results suggest that cryokarst features which dynamically develop during a retreat on the stagnating terminus substantially
speed up the length response and also noticeably reduce the volume response time. This is important for any long-term modelling studies involving debris-covered glaciers, as neglecting the effects of cryokarst results in an overestimation of transient response times during a warming phase. Furthermore, the resulting earlier and more enhanced mass loss rates agree better with the current observations of rapid thinning (Pellicciotti et al., 2015; Brun et al., 2018; Mölg et al., 2019) but the terminus response is still strongly delayed and requires warm periods of substantial durations (several centuries) to cause substantial
retreat (Fig. 9). This suggests that todays thinning may still be related to the warming after the Little Ice Age, or alternatively, it may be a consequence of our rather ad-hoc approach and threshold for the onset of cryokarst.

Numerous previous studies (Sakai et al., 2000, 2002; Benn et al., 2012; Buri et al., 2016; Kraaijenbrink et al., 2016; Miles et al., 2016; Ragettli et al., 2016; Watson et al., 2017; Rounce et al., 2018) have investigated the role of ice cliffs and supraglacial
ponds on the enhancement of melt on debris-covered glaciers and indicate some link between stagnation in ice dynamics and the development of such cryokarst features. Our model is, however, the first attempt to couple the effects of these features to glacier dynamics in a numerical model in order to explore the impact on glacier thinning. Although the ad-hoc approach used here is admittedly simplistic, it does allow for the general effect of cryokarst to be incorporated dynamically without requiring knowledge of the details of the physical processes, which are not yet completely worked out and would greatly complicate the

numerics since they are occurring at the sub-grid scale. The parameters chosen resulted in behaviour consistent with fractional area observations (Mölg et al., 2019; Steiner et al., 2019; Anderson et al., 2021, give a local area fraction up to 12%) and the approximate timing of the cryokarst evolution also matches the observation that stagnating glaciers tend to have more cryokarst (Benn et al., 2012).

The main limitation to this component of the model is that the choice of driving stress thresholds for the onset of cryokarst features is not well constrained by observations or directly linked to a sub-grid process based model. Hence it is clear that a better understanding of the link between glacier dynamics and the formation of cryokarst is needed. A more sophisticated model that faithfully represents the large scale, long-term effect of ice cliff and supraglacial pond evolution on the local surface mass balance would be useful for future studies.

**4.4   Transient response time**

The thinning and hence the volume response during retreat occurs in two distinct phases: a first relatively rapid response in the debris-free zone directly caused by enhanced melting, followed by a slower response in the debris-covered zone punctuated by the collapse of the stagnant terminus, and caused by the stagnation of the debris covered tongue. Although this has been indicated conceptually by general observations, Banerjee and Shankar (2013) gave the first dynamical explanation for this

behaviour using a simplified representation of the effects of debris cover. Here we use a more physically realistic model which includes debris evolution coupled to the ice flow and our results confirm their dynamical explanation. An important implication of this result, also pointed out by Banerjee and Shankar (2013), is that a simple volume response time scale to characterize the transient response of a glacier to climate forcing, developed for debris-free glaciers in Jóhannesson et al. (1989) and Harrison et al. (2001), does not seem possible for debris-covered glaciers because of its more complicated retreat behaviour.


To illustrate this, we calculate for our step change experiments (Sec. 3.2) the theoretical volume response time of Jóhannesson et al. (1989), which is given by

$$\tau_v = \frac{H_m}{-a_t}, \tag{11}$$

where $H_m$ is the maximum ice thickness and $a_t$ is the surface mass balance at the glacier terminus. It is, however, not that

clear how to define the terminal surface mass balance, as the glacier has both a debris-covered and a debris-free zone (frontal cliff) near the terminus. This is even more problematic when there is a zone of cryokarst at the terminus, so we neglect that case here. We choose four different terminus locations to extract the surface mass balance at the terminus from the modelling results and which depend on the location of extraction: for the response time $T_1$, the terminal surface mass balance is taken on the debris-free terminal ice cliff; for $T_2$, it is taken on the debris-covered zone just up-glacier from the ice cliff; for $T_3$, it is

taken as an average of the surface mass balances from $T_1$ and $T_2$; and for $T_4$, it is taken from the average over the last 300 m (or roughly ten ice thicknesses) including the debris-free ice cliff. The results of these response time calculations for both retreat and advance are found in Table 3. As is evident by comparison with the corresponding numerical volume response times, none

of these approaches gives reasonable theoretical predictions (Table 3) and results in either strongly over or underestimated response times, depending on whether the debris-free ice cliff is excluded. Note that using the theoretical volume response time of Harrison et al. (2001) does not make sense here, as this calculation takes into account the gradient of the surface mass balance near the terminus, which is close to zero wherever there is debris cover.

The presence and variability of debris cover brings into play additional dynamics that affect not only the volume response but also the geometry. The transient glacier thickness profile during a retreat showed two distinct shapes, depending on whether the stagnant and unsustainable tongue was still present. This time dependent glacier shape suggests that the volume–area power law scaling relationship that exists for debris-free glaciers (e.g. Bahr et al., 1997) is unlikely to exist in such a simple form for debris-covered glaciers. Volume–area scaling for debris-free glaciers, which rests on both theoretical arguments and observational data, shows that debris-free glaciers keep essentially the same shape even if they are not in steady state. This is clearly not true for the debris-covered glaciers modelled in our study.

Future work in establishing a way to understand and predict volume response times would be very beneficial here, as it would allow the approximate assessment of the large scale volume and length response to climate forcing without the need to run detailed, computationally expensive models for each glacier.

### 4.5 Steady state velocity–debris thickness relationship

The steady state profiles resulting from our model show an inverse relationship between debris thickness and ice flow velocity, consistent with both observations (Anderson and Anderson, 2018; Mölg et al., 2019) and numerical studies (Anderson and Anderson, 2016, 2018). It is natural to ask to what extent the debris thickness profile depends on the ice flow model and the debris transport model used. That question can be answered for the steady state case without assuming anything about the ice flow and considering only conservation of mass. In steady state and for the debris-covered domain, eqs. (1) and (3) can be written as one equation:

$$\kappa \frac{\partial(\bar{u}H)}{\partial x} + \frac{\partial(\bar{u}D)}{\partial x} = 0, \tag{12}$$

where $\kappa = cu_s/\bar{u}$. Integrating from the location of initial debris emergence (not necessarily at the ELA, as in our numerical model) to an arbitrary point $x$ further down glacier and rearranging, we obtain an expression for steady state debris thickness $D$ given by

$$D(x) = \frac{\kappa \bar{u}_e H_e}{\bar{u}} - \kappa H, \tag{13}$$

where $\bar{u}_e$ and $H_e$ are depth averaged ice velocity and ice thickness at the point of initial debris emergence, respectively. Near the terminus, the ice thickness $H(x)$ approaches zero and hence, the terminal debris thickness $D_{tr}$ can be expressed as

$$D_{tr} \simeq \frac{\kappa Q_e}{\bar{u}_{tr}}, \tag{14}$$

where $Q_e = \bar{u}_e H_e$ is the ice flux at the initial debris emergence point and $\bar{u}_{tr}$ is the ice velocity at the terminus. Note that we have not assumed SIA or any other ice flow model here, so there is no issue with vanishing velocity at the terminus, although even in that case one can require nonzero velocity at the terminus as in our debris transport model. Equation (14) is similar to equation (27) derived by Anderson and Anderson (2018) but with the difference that we have not assumed negligible englacial debris emergence and that our equation is only applied at the terminus.

Consistent with Anderson and Anderson (2018), Eq. (14) suggests that debris thickens towards the terminus as the velocity decreases. An interesting consequence of our formulation makes use of the fact that we have allowed for a variable velocity. In the case of a debris-covered glacier near steady state with an approximately uniform debris concentration, one can infer this concentration by measuring the representative velocity and debris thickness at the terminus and the velocity and ice thickness at the emergence location.

An additional feature of these model results is that for a fixed climate, the debris thickness profile in steady state appears to be approximately independent of concentration. Although this agreement is not perfect, one can see that the dashed lines indicating debris thickness in panels Fig. 2c and 2f are within 10% of each other for most of the ablation zone. For example, the two glaciers present at $x = 10$ km both have a debris layer of about 1 m thick even though their respective debris concentrations differ by a factor of two. The exception is the upper ablation zone just below the position of debris emergence (e.g. ELA), as there debris thicknesses are very low and relative differences therefore large. The differences in this zone seems to have a profound impact on the downstream velocity and ice flux gradient and hence seem to govern the final steady state glacier length. The very similar debris thicknesses for all concentrations in the lower part of the tongue directly imply almost identical surface mass balance at the same locations (for surface mass balance profiles see Fig S1).

## 4.6 Model limitations

We have used a well-tested but relatively simple ice flow model, the shallow ice approximation, which seemed to adequately capture the ice physics. Our debris transport model did not resolve the englacial transport of debris and assumed that all debris immediately starts melting out at the ELA. This approximation is a reasonable first order approach, especially for debris that is deposited onto the glacier surface in the accumulation zone near the effective ELA during transient simulations. This assumption works less well for glaciers whose debris deposition zone is far above the ELA, since the resulting emergence location will be located much further down glacier. It will likely not change any of our qualitative results but just shift the debris emergence location downstream from the ELA, resulting in the same general pattern of thickening of debris along the glacier and therefore essentially the same transient behaviour although at slightly different rates. It is also not an accurate method for determining the steady-state glacier extent for different climate scenarios, since due to the assumption of constant debris concentration, the total debris input into the system necessarily scales with the size of the accumulation area and accumulation rate. This is a significant shortcoming of the model, since it results in larger glaciers associated with colder climates possessing stronger debris source terms, which is not supported by observations (Banerjee and Wani, 2018). However, the general conclusions with

regard to transient behaviour and delay in terminus retreat of debris-covered glaciers still hold. Hence, our modelled tendency of advance or stagnation in a fluctuating climate is essentially a result of the asymmetric length response. For warm periods at time scales longer than the terminus retreat delay (e.g. several centuries) the random climate forcing experiments demonstrate the ability of the terminus to retreat substantially and to reset this memory to cold periods. Further, since the changes in total debris input stay in general relatively small and since the glacier does not exhibit a response until the additional debris has worked its way from the accumulation zone to the surface of the ablation zone, which can take decades to centuries, this becomes more important for final steady-state length and volume changes and is not expected to appreciably change the transient response of a debris-covered glacier far from equilibrium. To address this issue a model with the capability to track englacial debris transport would be required, which is beyond the scope of this study and is computationally much more expensive.

Another important issue is what happens at the terminus. In addition to the limitations of the cryokarst model discussed above, a further concern is the choice of boundary condition at the terminus. Our boundary condition is qualitatively similar to that used in Anderson and Anderson (2016), in that for both models there is a sub-grid scale rule for defining the interface between the debris-covered surface and an exposed ice terminus. The rare observations of advancing glaciers support the use of a terminal ice cliff (see Appendix A) but during retreat this is less commonly observed. Even so, the boundary condition we use captures the effects of a stagnating tongue and therefore still seems largely consistent with observations of terminus dynamics. More detailed observations of the termini or debris-covered glaciers and their effect on the glacier dynamics would be of benefit for future modelling studies.

## 5  Conclusions

We have presented a model that captures the essential processes governing debris-covered glacier dynamics while in a second step also integrating the effect of evolving cryokarst features on glacier evolution. Using this model, we have investigated the transient response of debris-covered glaciers to changes in climate. The results show that for a retreat the length response is strongly delayed compared to the volume response and that in general volume response times are much longer than for clean ice glaciers. This implies that periods of cold climate have a longer lasting effect on the transient volume and particularly on the length of debris-covered glaciers than do periods of warm climate. Such glaciers therefore tend to advance or stagnate in length in a fluctuating climate and hence glacier length is not representative of climate but rather depends on the history of cold phases. The modelled extended but generally stagnant glacier tongues in a warming climate are in agreement with observations. With regard to volume loss, the model is however much more responsive and can produce similar to observed substantial thinning and hence mass loss on the extended tongues due to stagnation or more specifically the cessation in local dynamic replacement of ice.

When cryokarst features are dynamically included in the model, it enhances both terminus thinning and the retreat rate and produces similar mass loss rates to those observed today. However, our cryokarst-model was rather simple and the related

parameters not well constrained, underscoring the need for a better understanding of the evolution of ice cliffs and supraglacial ponds so that they may be more accurately represented in long-term modelling studies.

Currently existing theoretical volume response times do not appear to be relevant for debris-covered glaciers because there is not a consistent way to define the surface mass balance at the terminus that gives theoretical values that match the numerical response times. Taking into account only the debris-covered area greatly overestimates the response time and this is likely due to the more complicated dynamics caused by the presence of the debris layer.

*Code availability.* Code used to generate the analyses used in this study is available at https://doi.org/10.5281/zenodo.4546678.

## Appendix A:  Boundary condition at the terminus

The choice of debris boundary condition near the terminus is not trivial, as many simple approaches lead to numerical simulations that exhibit unacceptable behaviours. Debris must leave the system before reaching the glacier terminus as otherwise the glacier can effectively grow without bound since debris may continue to thicken down-glacier indefinitely, thereby almost entirely insulating the glacier from melt. Hence, the debris boundary condition must be applied at a point up-glacier from the glacier terminus. However, boundary conditions that are defined at a location which depends on grid size, such as a fixed number of grid points from the glacier terminus, run the risk of exhibiting grid-size dependency in the numerical simulations. For example, we found that applying a debris flux condition one grid point from the glacier terminus results in a steady state glacier length that is heavily grid-size dependent, with the length varying by many times the mesh size. This is an undesirable outcome since the glacier physics should be independent of the numerical discretization used.

Observations of debris-covered glaciers that terminate in an exposed dynamically active ice cliff are numerous, occurring in relatively recent aerial images (e.g. Tsijiore Nouve, shown in Fig. A1a), early glaciological literature (Ogilvie, 1904), and even historical paintings (Escher von der Linth, 1794, as shown in Fig. A1b). While the terminus shown in Fig A1a is from a period of positive mass balance in the Alps (e.g. Mölg et al., 2019) and may represent an advancing glacier tongue, ice cliff termini have also been observed on several retreating debris-covered glaciers in the Himalaya (Evan Miles, personal communication, 2020). Motivated by these observations, we define the point at which the debris leaves the glacier to coincide with the location $x^*$ of a terminal ice cliff of critical thickness $H^*$, as shown in Fig. A2. The value of $x^*$ is found using a sub-grid linear interpolation on the grid points $x_j$ and the corresponding ice thicknesses at these grid points, $H_j = H(x_j)$. Hence $x^*$ is bounded by the grid points $x_i$ and $x_{i+1}$ such that $H(x_i) > H^*$ and $H(x_{i+1}) < H^*$, as shown in Fig A2. All the surface debris transported past the ice cliff location $x^*$ slides down the cliff and out of the system and is therefore removed from the surface by setting the debris thickness to zero for all grid points past $x^*$. The surface mass balance calculation at $x_{i+1}$ accounts for the sub-grid location of the ice cliff by using a weighted average of debris-covered and bare ice melt rates, with the weighting

dependent on location of $x^*$, and is given by:

$$a_{i+1} = a\frac{x^* - x_i}{\Delta x} + \tilde{a}\frac{x_{i+1} - x^*}{\Delta x}, \tag{A1}$$

where $a$ is the debris-covered surface mass balance, $\tilde{a}$ is the debris-free surface mass balance, and $\Delta x$ is the grid spacing.

Although the overall method is grid size dependent, it is convergent, i.e. the steady state glacier extent converges to a fixed value as the grid size is reduced. The results of convergence tests for the case of $c = 0.025\%$ for both warm, corresponding to ELA = 3100 m, and cold, corresponding to ELA = 3000 m, are shown in Table A1. The extents for the smallest grid size $\Delta x = 6.25$ m are used for the relative error calculations. We note that the relative error in steady state glacier extent for the

grid size used throughout the present study is roughly 5%, which is on the order of about 200 m. A plot of steady state glacier profiles corresponding to the tests shown in Table A1 can be found in supplementary figure S6.

A similar boundary condition was used in Anderson and Anderson (2016), where an ice cliff is also employed at the ter-

**Table A1.** Convergence test results for steady state glacier extent corresponding to $c = 0.025\%$.

| $\Delta x$ (m) | Extent (warm) (km) | Rel. error | Extent (cold) (km) | Rel. error |
|---|---|---|---|---|
| 100 | 5.86 | 0.281 | 9.66 | 0.241 |
| 50 | 6.97 | 0.145 | 11.17 | 0.122 |
| 25 | 7.71 | 0.055 | 12.13 | 0.046 |
| 12.5 | 7.99 | 0.021 | 12.48 | 0.019 |
| 6.25 | 8.16 | – | 12.72 | – |

minus and similar physics governs debris leaving the system, due to either tumbling down the terminal cliff or else by cliff

backwasting. One difference in their approach compared with ours is that debris covers the glacier right up until the terminus and leaves the system at the ice cliff at a prescribed rate, which for most simulations is set equal to the melt rate times the debris thickness. A second difference is the fact that the geometry of the terminal wedge appears to depend on the grid spacing, resulting in a changing geometry as the grid size is reduced. The implications of this for model convergence are not clear.

In contrast, the geometry terminal ice cliff used in the present study is essentially independent of grid size, since it depends on the critical ice thickness $H^*$. As the grid size is reduced, the position of the ice cliff is more accurately determined and therefore the terminus geometry converges to a steady value.

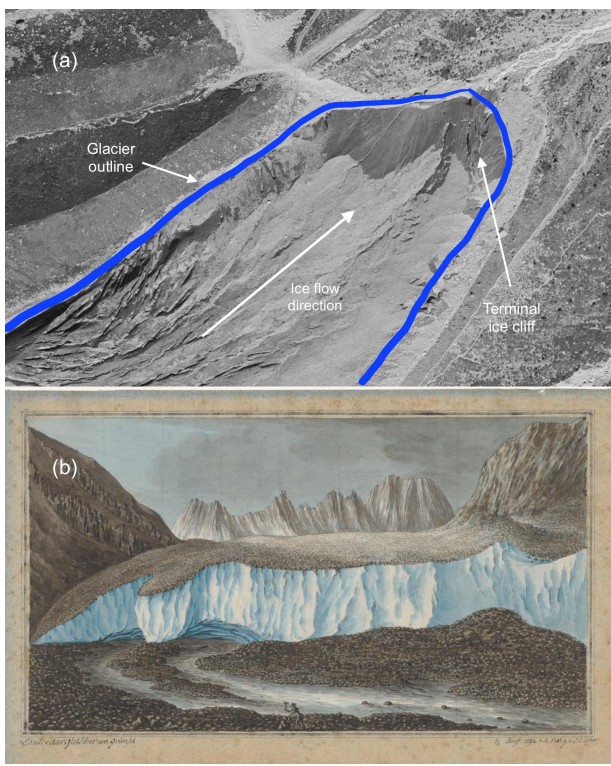

**Figure A1.** Examples of debris-covered glaciers with an ice cliff terminus: (a) satellite image of Tsijiore Nouve Glacier in 1988, reproduction with permission from Swisstopo (BA20059) and (b) a painting of the terminus of Unteraargletscher from 1794 (Escher von der Linth, 1794).

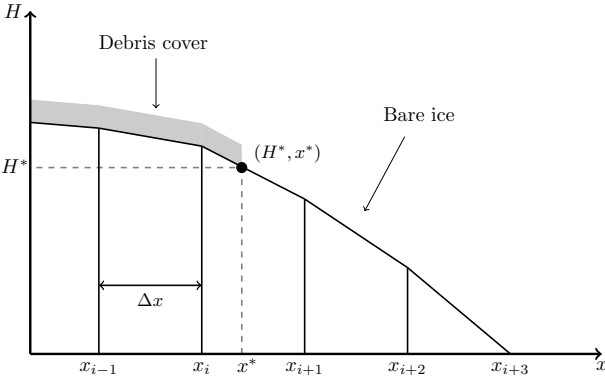

**Figure A2.** Schematic representation of the model terminus boundary condition. Debris covers the glacier only until a point $x^*$, corresponding to a critical glacier thickness $H^*$, and past this point the glacier is debris-free.

## Appendix B: Cryokarst model

The model coupling cryokarst features to glacier dynamics described in Section 2.1.5 requires threshold values of driving stress, $\tau_d^+$ and $\tau_d^-$, that define the presence of cryokarst near the terminus, which in turn reduces the insulating effect of the debris locally. We choose the values of the thresholds by examining the modelled driving stress of a debris-covered glacier during retreat and choosing threshold values that seem consistent with the onset of stagnation. The upper threshold between 100 and 125 kPa is consistent with driving stresses observed at the upper limit of the cryokarst zone at Zmuttgletscher in the Alps during its retreat from the Little Ice Age to today (100 to 130 kPa; Mölg et al., 2020). In Fig. B1a, we plot $\tau_d$ at 50 yr intervals during retreat following a step change between two steady states at ELA = 3000 m and ELA = 3100 m. In Fig. B1b and B1c, we show the corresponding glacier thickness $H$ and velocity $u$ during the retreat. Note that for driving stresses below the upper threshold (100 to 125 kPa) velocities drop to virtually zero (Fig B1c).

To illustrate the sensitivity of the model to variations in the thresholds, in Fig. B2 we rerun the experiments shown above in Fig. 6 but for different values of lower threshold $\tau_d^-$ while keeping constant $\tau_d^+ = 125$ kPa and $\lambda_m = 10\%$. For larger values of the lower threshold $\tau_d^-$, the full effect of cryokarst is felt sooner and therefore the volume and length response occur sooner (Fig. B2). However, the percentage of bare ice equivalent, shown in Fig. B2c, is less for larger $\tau_d^-$ values as the zone of stagnation driven cryokarst has less time to develop.

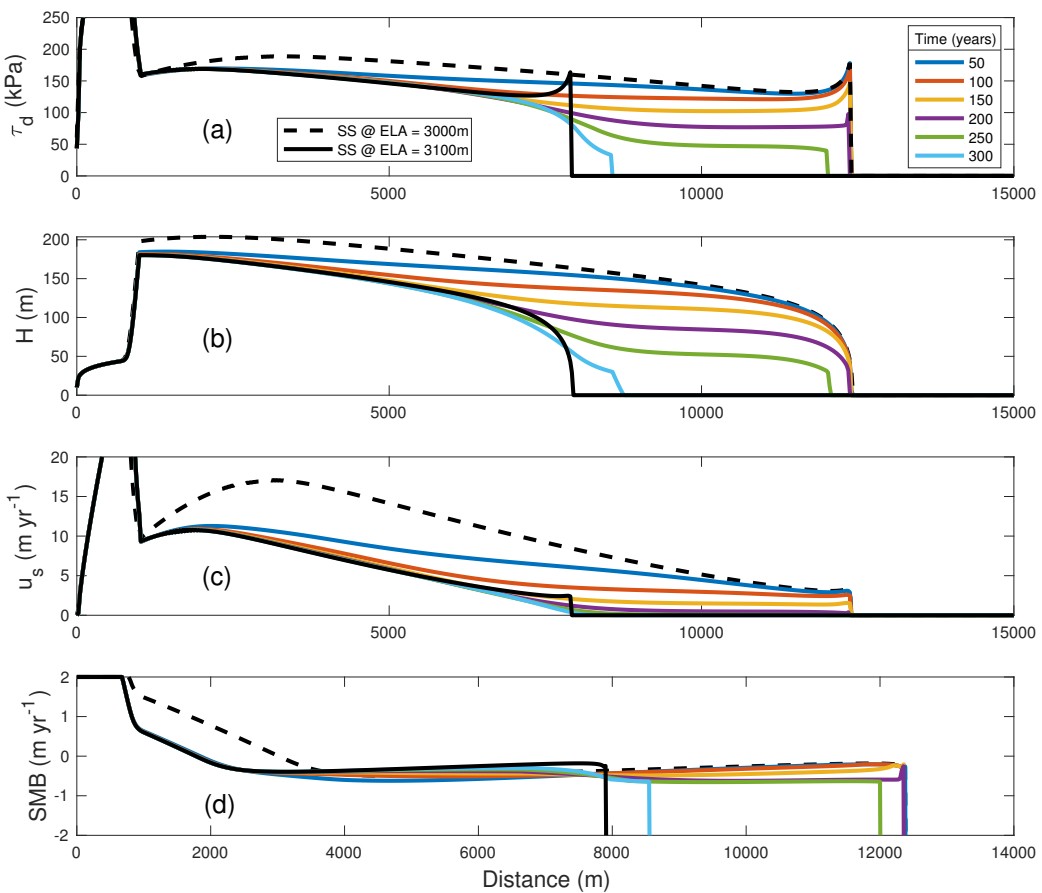

**Figure B1.** Effect of glacier retreat in response to step change in ELA from 3000 m to 3100 m on profiles of (a) driving stress $\tau_d$, (b) ice thickness $H$, surface velocity $u_s$, and surface mass balance at time intervals of 50 yr. The dashed black and solid black lines, respectively, refer to the initial and final steady states (SS) of the retreat experiment.

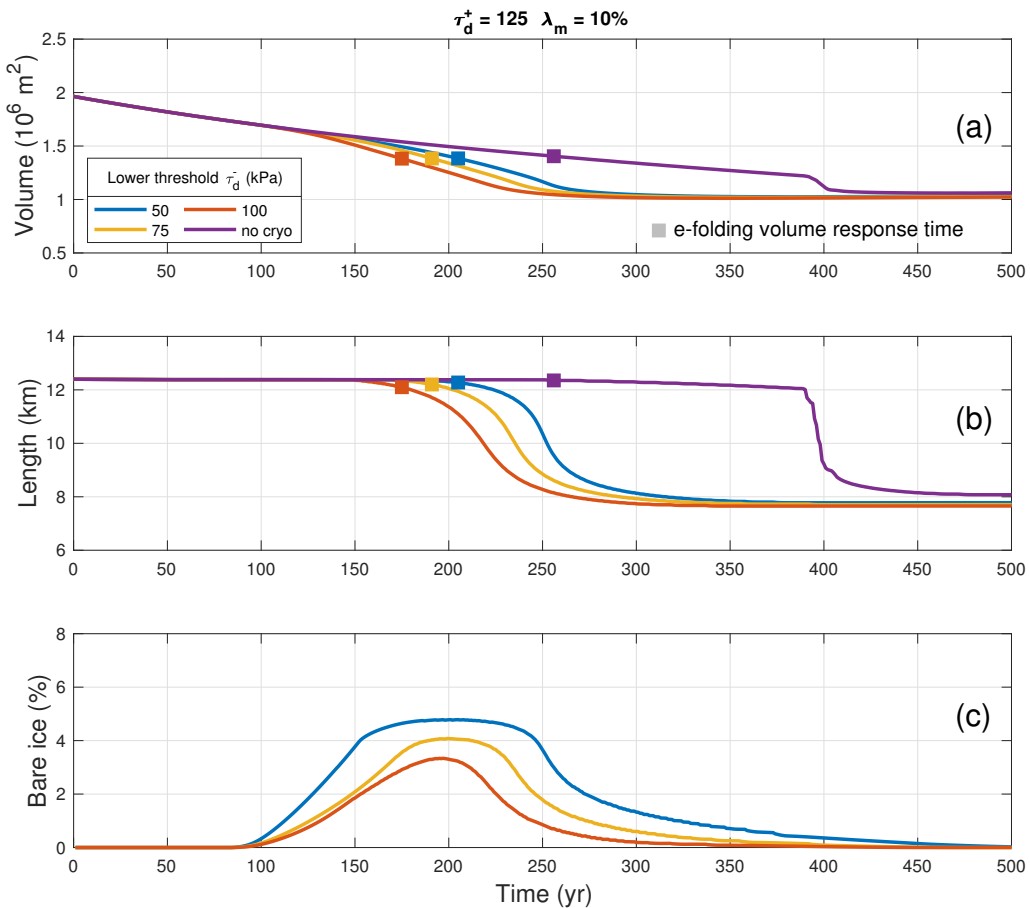

**Figure B2.** Sensitivity of model to different lower threshold values $\tau_d^-$ shown by coloured lines for (a) transient volume response, (b) transient length response and (c) bare ice equivalent debris-covered area during a retreat after a 100 m step change in ELA from 3000 m to 3100 m. In all cases, the debris concentration is $c = 0.25\%$, the upper driving stress threshold is $\tau_d^+ = 125$ kPa, and the maximum cryokarst fraction is $\lambda_m = 10\%$.

*Author contributions.* JF and AV designed the study. JF wrote the code and performed the numerical experiments. JF interpreted the data and wrote the paper with input from AV.

*Competing interests.* None.

*Acknowledgements.* We thank Tobias Bolch and Nico Mölg for helpful discussions. We also thank the scientific editor, Harry Zekollari, and two reviewers, Leif Anderson and Fabien Maussion, for numerous useful suggestions that greatly improved the quality of the final manuscript. This research was financially supported by the Swiss National Science Foundation grant number 169775, entitled "Understanding and quantifying the transient dynamics and evolution of debris-covered glaciers".

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
