# Peer review of "Modelling steady states and the transient response of debris-covered glaciers"

_The Cryosphere, 2020_

## Referee Comment (RC1) · Leif Anderson (Referee) · 23 Nov 2020

Review of "Modelling steady states and the transient response of debris-covered glaciers"

by Leif Anderson

The manuscript develops our understanding of debris-covered glacier response to climate change using a numerical model run in an idealized setting. A vital new contribution is a parameterization of melt hotspots on debris-covered glacier evolution and response time. The manuscript also describes the long-term response of debris-covered glaciers to a variable climate.

**Major comments**

This is a welcome and important effort that expands our understanding of debris-covered glacier evolution! Overall, I think that this manuscript will be a great/strong contribution once the below issues are addressed. Many of these comments are related to presentation, but they are vitally important so this work can be easily understood by TC audience.

This work is emerging from a dialogue between other DCG modeling and observational efforts. As the paper reads now, that dialogue is not yet properly developed. Often times previous work immediately relevant to points being developed in this manuscript are cited (or not) as having worked in the general topic at the start of a paragraph. But the insights gained from past efforts are not yet allowed to be in dialogue with the results from this work.

This partly means that a stronger foundation should be laid in the manuscript (in the introduction) with regards to what insight has already been gained from previous work and how this effort builds off of those previous efforts. This also means that the writing does not clearly delineate between conclusions made by previous work and the new findings here (especially in the discussion section and the toe parameterization appendix). I raise this point not to diminish the important contributions made here. On the contrary engaging past work with the new insights will highlight the work done here more clearly and make for an even more valuable contribution to the community.

Because the model developed in Anderson and Anderson (2016) and the one presented here very similar I think it is appropriate to be a bit more explicit about how the models are different. As it reads now it is not always clear what was originally derived by A & A 2016 and what is new here. A bit more care should be taken when discussing the differences between the toe parameterization approaches. It is unclear how different the approach derived here is different from the range of parameterizations explored in A & A 2016. A more explicit statements about the toe method will allow the method developed here to be reproduced.

Figures in general are well composed, though a few more simple figures will expand accessibility to a broader audience. It would be helpful to see some modeled mass balance profiles plotted in the main paper since they are essential for showing the difference between debris-free and debris-covered glaciers. And the added effect of cryokarst on DCG mass balance profiles.

A figure or schematic showing how the cryokarst formulation is implemented in the model would be helpful. Maybe driving stress could be plotted and an example of the effect of the cryokarst features on the mass balance profile would aid the reader in understanding the new parameterization and its effect on the glacier. As the manuscript is now I have trouble visualizing the pattern of cryokarst features on the glacier at any one time. I ultimately think this parameterization is an important and useful contribution and showing a bit more how it works will only benefit the manuscript and the community. This is an important contribution! The authors might also consider

shifting from the use of 'cryokarst,' as ice cliffs themselves are not necessarily the result of the collapse of englacial tunnels.

The authors might consider adjusting the use of the term 'white noise' as it refers to the climate forcing. In terms of climate, white noise forcing almost always refers to year-to-year variability in the climate. The climate forcing applied here is actually red noise because the timestep is 100 years and there is therefore autocorrelation from year-to-year. This manuscript uses persistent climate changes to fore the model. I am not actually sure of the correct phrasing but maybe climate changes that are randomly sampled from a normal distribution would interface better with previous work. Or just the response of a DCG to variable climate?

The discussion section would be improved with a more thorough discussion of the uniform englacial debris concentration assumption. It is very important to consider what a steady, uniform englacial debris concentration implies for headwall erosion rates when glacier geometry is changing. I have included/expanded on some points lower down.

The manuscript in general should be streamlined and repeated statements should be cut out. Individual sentences are well composed, but I find myself a bit overwhelmed at times in the text. The modeling results section will benefit the most from some textual work. The number of experiments and the changing focus from various parts of the DCG system make it hard to follow. Anything that can be done to simplify and distill the description of these experiments will help the reader.

**Line-by-line comments**

Line 13. "as is also observed in remote sensing" this could be a little clearer. Maybe just remove 'in remote sensing'

line 40. "the relatively recent advent of remote sensing data." consider re-phrasing here.

44. The introduction is a bit parsimonious towards previous efforts. What are the contributions of previous debris-covered glacier models? What have we learn up to now? By setting the stage more the novel and interesting contributions of this work, which there are many, will be better highlighted.

Line 46 and 47. Recognizing that you have cited several of our papers here, but there are additional transient simulations of debris-covered glaciers responding to climate change using essentially the same model as A &A 2016 in Crump et al., 2017 and Anderson et al., 2018. The references are fully written at the end of the manuscript. Also Anderson et al., 2019a does not include any model simulations.

Line 57-59. The way this sentence is written it is fuzzy what the actual differences are between the models. Is it the same besides the differences listed here? If not it would be good to make it a bit more clear what the other differences are in the methods section.

Line 63-65. This is a great way to support the use of SIA!

Line 90. Reading this sentence makes it seem like this melt formulation (equation 6) was derived by Nicholson and Benn (2006) but it was actually derived in Anderson and Anderson (2016). It is appropriate to cite that work here.

Line 99-100. How is it different? It seems to be nearly the same. It might be more appropriate to state that you 'improve upon' or 'start from.' How is this different from Anderson and Anderson (2016)? Explicitly stating what they do will make it more clear what the new contributions of this work are.

Also how was your value of $D_0$ chosen?

Line 102. This sentence could be simplified right now it is a little more complicated than it needs to be.

Lines 103-105. Your case would be stronger if you develop the justification for the parameterization a bit better here. It seems to me that there are some more citations here for work that has linked ice flow with these features. Like Kraaijenbrink et al., 2016 and/or Watson et al., 2017. It is a clever approach though.

Lines 105-110. It would be helpful for the reader to include the equation for driving stress here. That way readers can connect to the fact that driving stress scales with ice thickness and surface slope. Is there a physical mechanism why cryokarst features might follow driving stress? Would be good to include that.

Line 122. Just need to clarify what the CFL condition is here as this is the first time this acronym shows up in the text.

**Section 3 Modelling results**

This section is rather difficult to follow and I am quickly overwhelmed by the number of simulations and how quickly the writing moves between them.

132-135. It could be beneficial to include a what you refer to as a 'baseline' case (with base debris concentration) so the reader has a single simulation to compare the others to. Reading below it is easy to get lost in all of the simulations. Maybe this baseline case could be bolded in the figures below?

Figure 1. Nice figure. What part of the glacier is covered with debris? How does that relate to the ELA. Perhaps adding these would be helpful to bring the various components of the model together for the reader.

Line 157 need a hyphen between 'debris' and 'covered'

Section 3.2 I think these are all important interesting simulations. This section would be improved though with a bit more synthesis. It is a bit difficult to follow because of the number of different experiments. Maybe more clear topic sentences clearly keying on what each experiment the paragraphs correspond to would help? Or sub-section titles for each experiment?

It might also be that the description moves between simulations using different englacial debris concentrations quickly. Perhaps it would be easier to follow if the descriptions of the experiments use one concentration case?

Figure 2 is really a great future!

223-224. Might be good to have a citation here.

Figure 4 is also really clear.

Table 4. The table looks very clean but maybe adding in text at the top the definition of each variable again would help the reader follow.

Figure 6. The introduction of 'Bare ice %' is hard to wrap my head around since it seems to be a new way of describing the cryokarst features. Maybe just label it % of the surface composed of cryokarst features. Consider finding another way to represent the contribution of cryokarst that is more clear.

Figure 7. You might consider moving this figure into the supplemental and just describing the effect of cryokarst on long term evolution in the text. Just so the reader does not feel overwhelmed.

**4.1 Debris-covered glacier memory**

This section highlights some interesting findings. The section, though, would be improved by stating what past studies have concluded related to this topic and then emphasizing showing how your results/conclusions differ. This is especially relevant to interface a bit with past transient glacier model simulations. Do they show a similar effect that support your discussion here?

299-302. This paragraph would benefit from a look at the past literature on the subject, as this point has been raised previously. Additionally Clark et al., 1994 et al. also discuss this effect.

Line 347 -351. There are studies that do connect ice cliff occurrence to ice dynamics, including Benn et al., 2012; Kraaijenbrink et al., 2016, .

**Section 4.4 Steady state velocity–debris thickness relationship**

I think this is a very interesting section. I do think it would be improved if it interfaced with the previous literature on the topic. Especially emphasizing how this work has expanded on those previous insights.

392-393. A & A 2018 also do a compilation of 8-10 glaciers that show that debris thickness patterns follow this same pattern. These observed profiles can also be referenced with the Mölg study as well.

395. "It is natural to ask to what extent the debris thickness profile depends on the ice flow model and the debris transport model used. That question can be answered for the steady state case without assuming anything about the ice flow and considering only
conservation of mass."

It is unclear how the statement above relates to the rest of the paragraph. This seems like an interesting topic though.

Equation 11 is very similar to one derived by Anderson and Anderson (2018) who follow a similar approach. It seems appropriate to cite that you are following that line of logic or interface with that work here.

404. How is it possible that there is ice flow at the terminus that is not 0? The SIA is based solely on internal deformation which is requires that ice thickness is larger than 0 which is not the case at the terminus. Just a bit of clarification will help.

409. There is an interesting discussion to be had between the insights from A &A 2018 (Fig. 9) and what is discussed in this paragraph, especially regarding the zone of englacial debris emergence as described there. How does this discussion mesh/build off of with what was discussed in A &A 2018?

**4.5 Model limitations**

421-425. This is a repeat from a point made above. My sense is that this only needs to be stated once.

426 to 431. The authors should discuss the implications of the assumption of uniform englacial debris concentration further.  From my view it seems more fair to say 'that the effect of a uniform englacial debris concentration should be explored further.' I mention this because there are a number of simplifications that go into this assumption.

I think its is a reasonable first order approach, but this means the entire ablation area will be covered with debris.

It needs to be added here that different ice flow paths will change the englacial debris concentration even with a uniform input of debris everywhere on the glacier. It is really impossible to have a glacier with a uniform englacial debris concentration because of the straining of ice and the inevitable variability of debris input (in space and time) to the glacier surface.

One additional point that should be discussed is how applying a uniform englacial debris concentration relates to headwall erosion rates. If the headwall erosion rate is constant in time then as a glacier gets bigger the englacial debris concentration by definition must become smaller.

This effect is not included in this model. By keeping the englacial debris concentration uniform and steady there must be a requisite increase in headwall erosion rate as the glacier grows in size. If the glacier doubles in size then the headwall erosion rate would need to also double. I think this is simply an underlying assumption of this approach that should be clear to the reader and if possible should be quantified and placed in the supplemental material.

427. missing period.

434. It should be made explicitly clear what the differences are between the toe condition applied here and the one presented in A and A (2016) in the main text. Is it simply a modification of the approach presented in A and A 2016? Are they not also quantitatively similar? See the text regarding the toe parameterization in the Appendix below.

436-437. I could not find where this statement is discussed in the appendix. Is there a citation that notes this or is it a new observation? I am unaware of this effect.

437-438. The way this paragraph is written implies that A &A 2016's approach would not capture the effects of a stagnating tongue. Is this actually true? Looking at the other publications after A &A 2016 like Crump et al., 2017 and Anderson et al., 2018 the length change curves are similar to those presented here and based on the toe parameterizations the dynamics should be represented similarly to the work here.

**Appendix A and the toe parameterization in general**

It is a substantial effort to develop a toe parameterization and any improvements on the exploration from A & A 2016 are welcome, important, and vital for the future development of debris-covered glacier models. It is also important that the method presented here also be reproducible. It would be good for the authors to describe the sub-grid interpolation scheme in detail. What shape/formula do you assume? What $H_*$ terms are viable?

465-475. It seems like there should also be some discussion of how this formulation relates to the original terminal condition described by A & A 2016. How are the approaches different?

A & A 2016 explore a range of possibilities for the terminal parameterization which the toe is drowned in debris because it cannot leave the glacier and also a case in which an ice cliff persists at the terminus and debris is effectively rapidly removed from the glacier. See Figure B1 and section 5.2 in A & A 2016. Ultimately, A & A 2016 use a scheme where debris is removed based on the bare ice melt rate which is basically the same as is implemented here. I think it would benefit the readership to have a more complete description of the differences between the two schemes and how different they actually are.

It seems that the parameterization presented here is a smart approach. Despite the way the text is written it seems the approach follows the A & A 2016 formulation closely and fits almost within the range of parameters explored there. The new approach presented here essentially sets no limit on the d_flux term from A & A 2016, and the formulation presented here would be close to the c =10 case in Figure B1 from A & A 2016 for debris removal. The main difference is that this approach keeps the ice cliff backwasting at the bare ice melt rate despite the removal of more debris than that backwasting of the ice cliff actually would allow.

The down side of the approach presented here is that the removal of debris from the glacier is not necessarily physically representative of the process of debris removal at the terminus of real debris-covered glaciers.

The A & A 2016 scheme honors that the removal of debris from the toe in the ice cliff case is determined by the backwasting rate, but this in turn leads to a greater grid scale dependence than the scheme presented here. From my view the benefit of either of these schemes depends on the decision to value either grid-scale dependence or the physical representativeness of debris removal from the toe.

Either way a more nuanced description of this toe scheme and how it relates to the work of A & A 2016 is needed to ensure the community can follow these methodological differences.

Figure B1. It seems like this figure should plot the mass balance curve with time, since the the cryokarst parameterization adjusts that directly. I would also like this figure with the SMB curves included in the main manuscript since the cryokarst parameterization is a central, new contribution of this work.

References

Anderson, R. S., Anderson, L. S., Armstrong, W. H., Rossi, M. W. and Crump, S. E.: Glaciation of alpine valleys: The glacier – debris-covered glacier – rock glacier continuum, Geomorphology, 311, 127–142, doi:10.1016/j.geomorph.2018.03.015, 2018.

Crump, S. E., Anderson, L. S., Miller, G. H. and Anderson, R. S.: Interpreting exposure ages from ice-cored moraines: a Neoglacial case study on Baffin Island, Arctic Canada, Journal of Quaternary Science, 32(8), 1049–1062, doi:10.1002/jqs.2979, 2017.

Kraaijenbrink, P. D. A., Shea, J. M., Pellicciotti, F., Jong, S. M. de and Immerzeel, W. W.: Object-based analysis of unmanned aerial vehicle imagery to map and characterise surface features on a debris-covered glacier, Remote Sensing of Environment, 186, 581–595, doi:10.1016/j.rse.2016.09.013, 2016.

Watson, C. S., Quincey, D. J., Carrivick, J. L. and Smith, M. W.: Ice cliff dynamics in the Everest region of the Central Himalaya, Geomorphology, 278, 238–251, doi:10.1016/j.geomorph.2016.11.017, 2017.

---

## Referee Comment (RC2) · Fabien Maussion (Referee) · 29 Nov 2020

In this study, the authors apply a debris cover parameterisation to a flowline model in order to study some aspects of the transient response of debris covered glaciers to climate change. This paper helps to better understand the dynamics of debris covered glaciers, and I very much enjoyed reading it (I may be biased because I like idealized numerical studies).

Parts of my review originate from a discussion with colleagues Lilian Schuster and Lindsey Nicholson, and I would like to acknowledge their contribution here. Finally, I would like to apologize for my late review which doesn't serve well this nice paper.

[Figure]

**1 General comments**

- At the end of the introduction, you write: "*to date no study has used a coupled ice flow-debris transport model to study in detail the transient response and characteristic response times of a debris-covered glacier. This study aims to fill this gap...*". "It has never been done before" is not a good motivation for a study, and I think that the paper would gain from clearly stated research questions. In particular, it would help to understand what motivated the model design and the design of the idealized experiments (why this bed profile, why this model design, etc.). Research questions will also help to place the study in the context of previous literature, and prepare the reader to understand what you are trying to achieve with this paper.

- The word "idealized" does not show up in the title, abstract, or introduction. I think it should be clearly stated much earlier (maybe not in the title, but at least in the abstract). "Numerical modeling" could be understood as "applied to real glaciers".

- This may be subjective, but I don't find any of the comparisons with Jóhannesson's response times informative or useful. Even without debris cover, you can find numerical response times of glaciers which are widely different than the analytical ones, since the e-folding times are highly dependent on parameters such as bed depressions, mass-balance (MB) gradients, etc. (see e.g. Zekollari et al. 2015 or Schuster, 2020 - unpublished thesis work).

- Your code availability statement ("available upon request") is against this journal's data and open science policies: https://www.the-cryosphere.net/policies/data_policy.html. I strongly recommend to make your code available (under a clear license), which will increase the visibility and re-usability of your work.

**2 Specific comments**

**Abstract** I'm not very familiar with the debris-covered glacier literature, but I had to search for "cryokarst" online

**Abstract** add "idealized numerical simulations"

**eq. (1)** consider using $b$ instead of $a$ for mass-balance (more common I believe)

**L72** "for a given a bed elevation" - remove "a"

**L96** having read section 2.1.4 and the appendix, it's still not clear to me how you compute $H^*$ (and I don't want to check up on Anderson et al 2016). I notice later that $H^*$ is a constant and a model parameter: mention this earlier in the text.

**L100** specify which appendix.

**Appendix A** despite of your valid attempts to show that this boundary condition may be found in the real-world, I still believ that the ice-free terminus condition is more a model necessity (trick) than a real-world feature. You don't have to change anything in the text here, I just wanted to comment on that.

**Sect. 3.1 (steady states)** I really had to think twice about how you can reach steady state with such a model. I think that it would help to write more about it. E.g. by saying again that (i) steady state can be reached only because the MB doesn't go too close to 0 and (ii) that this is only possible by removing debris at the terminus and effectively capping the debris thickness to a reasonable value. You can refer to Fig. S1 in this section (or mention typical values of MB at the terminus in the model) to help understanding.

**L155** to our knowledge, Jóhannesson et al, (1989) wrote:"The volume time-scale tau can be computed from the volume differences between two steady-state profiles

scaled to the causal mass- balance change", but did not mention the e-folding volume response time (yet). Maybe refer to another paper as well: e.g. Oerlemans (1997) or Jóhannesson (1997)

**L185** volume-area scaling: since it might be unclear to some of your readers, add here that (in your model) area is directly proportional to length

**L190** is "stagnant" the correct term here? I was confused several times in the manuscript about this, because you seem to use "stagnant" for when the glacier length does not change. Personally, I understand "stagnant" as "ice that is not moving" ($u = 0$). You cannot have "stagnant" ice with your numerical model setup. I would argue for using "stable terminus" in place of "stagnant", or clearly state in the text what you mean with "stagnant". At the very end of the paper there is a sentence going in this direction ("*stagnation or more specifically the cessation in local dynamic replacement of ice.*").

**L277** "stagnated": same here. Is it the correct way to say that? Non-divergence is still happening with $u$ constant and non-zero, i.e. moving ice.

**Section 3.3 "white noise"** traditionally, white noise climate should be applied on a year to year basis and the periods of cold and warm climates would occur "naturally", as a result of random sampling. I wonder how this would affect your results. Additionally, I wonder if an annually varying MB would still work with your debris cover formulation, since you don't deal with temporary ice/snow cover on debris as for now.

**Fig. 5** while this figure carries well your main message, I think that it can be misinterpreted. In particular, the blue line in Fig. 5b gives the impression that the glacier will always grow, i.e. never reach a "steady state" (i.e. an average length around which it oscillates - albeit in a strange, debris covered way). What you could do here is continue the simulation for an additional 5k years (at least) and see what

happens. It might have an interesting consequence: the "average length" of a debris covered glacier under a *random* climate might be longer than the length of the same glacier under *constant* forcing. I expect the average length to be somewhere between the steady state lengths with the two ELAs (although it might even be longer than that, which would be very interesting to discuss further).

**Figure S1** : write that "SS" stands for "steady state" in the legend.

**3  References**

Johannesson, T.: The response of two Icelandic glaciers to climatic warming computed with a degree-day glacier mass-balance model coupled to a dynamic glacier model, J. Glaciol., 43(144), 321–327, doi:10.1017/S0022143000003270, 1997.

Oerlemans, J.: A flowline model for Nigardsbreen, Norway: Projection of future glacier length based on dynamic calibration with the historic record, Ann. Glaciol., 24, 382–389, doi:10.1017/S0260305500012489, 1997.

Schuster, L.: Response time sensitivity of glaciers using the Open Global Glacier Model : From idealised experiments to an estimate for Alpine glaciers, Universität Innsbruck. [online] Available from: https://bibsearch.uibk.ac.at/AC15655181, 2020.

Zekollari, H. and Huybrechts, P.: On the climate–geometry imbalance, response time and volume–area scaling of an alpine glacier: insights from a 3-D flow model applied to Vadret da Morteratsch, Switzerland, Ann. Glaciol., 56(70), 51–62, doi:10.3189/2015AoG70A921, 2015.

---

## Author Comment (AC1) · 23 Dec 2020

**Modelling steady states and the transient response of debris-covered glaciers**

James Ferguson and Andreas Vieli

Submitted for review to The Cryosphere (https://doi.org/10.5194/tc-2020-228)

**Author Response to Both Reviews**

We would like to thank both reviewers for two very helpful reviews. We believe their comments can be easily addressed via the intended revisions outlined below. We first discuss major themes and then go line by line, first for Leif Anderson's review (starting on page 2) and then for Fabien Maussion's review (starting on page 11).

**Major Themes**

**C1 Dialogue, literature, foundations**

It was raised by both reviewers that we could improve the manuscript by having a better discussion and integration of the advances made by previous work, what research gaps are present, and where our work fits into this dialogue with other existing research. We agree that our paper would gain from a better discussion in context of literature. We will include the additional references suggested by the referees and beyond and better incorporate these into the existing text. We will also use this to better motivate our study and model choice.

**C2 Model description**

The reviewers felt that some aspects of our model were not so clearly explained or motivated. We will improve the explanation and details, explain why the features of this model were chosen, and what is different from other models used in previous studies. We will have a particular focus on differentiating the relevant parts of our model (e.g. frontal boundary condition) from that used in Anderson and Anderson (2016) and on the aspects of our cryokarst model.

**C3 Additional figures to aid visualization**

There were a few places where additional figures could be useful for making our results easier to understand. We agree with these points and we will add such figures where noted (e.g. visualize impact of cryokarst effect on along-flow surface mass balance). In order to better illustrate the role of the debris for the transient response, we further plan to add animations of some of the modelling experiments in the supplements.

**C4 Variable climate forcing**

We used 'white noise forcing' in the text and both reviewers felt this was incorrect. We agreed that the terminology we used was not fully correct. We will note that it is white noise on a century timescale and we will use the term 'random climate forcing' instead.

**C5 Uniform debris concentration assumption**

One reviewer makes an important point about the assumptions inherent in our choice of constant debris concentration. We did not yet discuss the impact of these assumptions and limitations very clearly in the text. But we feel this issue needs more attention and is especially important given that numerous other studies use the same assumption. Therefore, we will add a clearer and more substantial discussion of the impact and limitations of the assumptions of this model choice. However, based on our existing results we are confident that this assumption does not affect the main conclusions of our research.

*In what follows, we show first the reviewer's comments in black and then our response in blue.*

**Major comments (Reviewer 1: Leif Anderson)**

This work is emerging from a dialogue between other DCG modeling and observational efforts. As the paper reads now, that dialogue is not yet properly developed. Often times previous work immediately relevant to points being developed in this manuscript are cited (or not) as having worked in the general topic at the start of a paragraph. But the insights gained from past efforts are not yet allowed to be in dialogue with the results from this work.

We will better incorporate the previous work into the manuscript both for context and to explain what is new (and what isn't) in this paper.

This partly means that a stronger foundation should be laid in the manuscript (in the introduction) with regards to what insight has already been gained from previous work and how this effort builds off of those previous efforts. This also means that the writing does not clearly delineate between conclusions made by previous work and the new findings here (especially in the discussion section and the toe parameterization appendix). I raise this point not to diminish the important contributions made here. On the contrary engaging past work with the new insights will highlight the work done here more clearly and make for an even more valuable contribution to the community.

Because the model developed in Anderson and Anderson (2016) and the one presented here very similar I think it is appropriate to be a bit more explicit about how the models are different. As it reads now it is not always clear what was originally derived by A & A 2016 and what is new here. A bit more care should be taken when discussing the differences between the toe parameterization approaches. It is unclear how different the approach derived here is different from the range of parameterizations explored in A & A

2016. A more explicit statements about the toe method will allow the method developed here to be reproduced.

We will clarify the description of model, and in particular expand and better explain and justify both the terminal boundary condition and the cryokarst parameterization.

Figures in general are well composed, though a few more simple figures will expand accessibility to a broader audience. It would be helpful to see some modeled mass balance profiles plotted in the main paper since they are essential for showing the difference between debris-free and debris- covered glaciers. And the added effect of cryokarst on DCG mass balance profiles.

We will include surface mass balance plots in the manuscript (for the cryokarst case also) and add some animations in the supplements for illustrating the differences in transient response.

A figure or schematic showing how the cryokarst formulation is implemented in the model would be helpful. Maybe driving stress could be plotted and an example of the effect of the cryokarst features on the mass balance profile would aid the reader in understanding the new parameterization and its effect on the glacier. As the manuscript is now I have trouble visualizing the pattern of cryokarst features on the glacier at any one time. I ultimately think this parameterization is an important and useful contribution and showing a bit more how it works will only benefit the manuscript and the community. This is an important contribution! The authors might also consider shifting from the use of 'cryokarst,' as ice cliffs themselves are not necessarily the result of the collapse of englacial tunnels.

We already plot the driving stress in the appendix and in the supplement, but we will try to visualize the thresholds in driving stress as well and try to add a sketch for explaining the cryokarst parameterization.

The authors might consider adjusting the use of the term 'white noise' as it refers to the climate forcing. In terms of climate, white noise forcing almost always refers to year-to-year variability in the climate. The climate forcing applied here is actually red noise because the timestep is 100 years and there is therefore autocorrelation from year-to-year. This manuscript uses persistent climate changes to fore the model. I am not actually sure of the correct phrasing but maybe climate changes that are randomly sampled from a normal distribution would interface better with previous work. Or just the response of a DCG to variable climate?

We agree, as noted above in main response C4. We will use 'random climate forcing' instead.

The discussion section would be improved with a more thorough discussion of the uniform englacial debris concentration assumption. It is very important to consider what a steady, uniform englacial debris concentration implies for headwall erosion rates when glacier geometry is changing. I have included/expanded on some points lower down.

We agree, as noted in main response point C5 above. We will note this in the manuscript and be more explicit about how this pertains to the debris input and discuss the implications on our findings. Based on our existing modelling results we are however

confident that our main conclusions with regard to general dynamical behavior of debris covered glaciers (in contrast to clean ice glaciers) remain unchanged.

The manuscript in general should be streamlined and repeated statements should be cut out. Individual sentences are well composed, but I find myself a bit overwhelmed at times in the text. The modeling results section will benefit the most from some textual work. The number of experiments and the changing focus from various parts of the DCG system make it hard to follow. Anything that can be done to simplify and distill the description of these experiments will help the reader.

We will better streamline the text and try to avoid repetitions.

**Line-by-line comments (Reviewer 1: Leif Anderson)**

Line 13. "as is also observed in remote sensing" this could be a little clearer. Maybe just remove 'in remote sensing'

Agreed – will remove "in remote sensing".

line 40. "the relatively recent advent of remote sensing data." consider re-phrasing here.

Agreed – will rephrase.

44. The introduction is a bit parsimonious towards previous efforts. What are the contributions of previous debris-covered glacier models? What have we learn up to now? By setting the stage more the novel and interesting contributions of this work, which there are many, will be better highlighted.

Agreed – as mentioned in the main comment C1, we will explain in detail what was done previously, what the state of the field is up until now, and be more explicit about what is new in our work/model.

Line 46 and 47. Recognizing that you have cited several of our papers here, but there are additional transient simulations of debris-covered glaciers responding to climate change using essentially the same model as A &A 2016 in Crump et al., 2017 and Anderson et al., 2018. The references are fully written at the end of the manuscript. Also Anderson et al., 2019a does not include any model simulations.

Good point – as mentioned in C1, we will add / correct the references here.

Line 57-59. The way this sentence is written it is fuzzy what the actual differences are between the models. Is it the same besides the differences listed here? If not it would be good to make it a bit more clear what the other differences are in the methods section.

Agreed – as mentioned in C2, we will give more detail on our model and what is different from previous models.

Line 63-65. This is a great way to support the use of SIA!

Thanks.

Line 90. Reading this sentence makes it seem like this melt formulation (equation 6) was derived by Nicholson and Benn (2006) but it was actually derived in Anderson and Anderson (2016). It is appropriate to cite that work here.

Good point – we will change to reflect this.

Line 99-100. How is it different? It seems to be nearly the same. It might be more appropriate to state that you 'improve upon' or 'start from.' How is this different from Anderson and Anderson (2016)? Explicitly stating what they do will make it more clear what the new contributions of this work are.

As discussed in main comment C2, we will address this by giving more details on our model and how it differs from previous work in both the Methods section and in the appendix.

Also how was your value of $D_0$ chosen?

We will address this also in the Methods.

Line 102. This sentence could be simplified right now it is a little more complicated than it needs to be.

Agreed – we will rephrase.

Lines 103-105. Your case would be stronger if you develop the justification for the parameterization a bit better here. It seems to me that there are some more citations here for work that has linked ice flow with these features. Like Kraaijenbrink et al., 2016 and/or Watson et al., 2017. It is a clever approach though.

As mentioned in main comment C2, we will add more details on the model here. We will look more carefully at these references as well to see if it makes sense to cite them. As there are many studies on ice cliffs / ponding and some qualitative but somewhat vague relation of these features to dynamics. However, to our knowledge there are currently no explicit quantitative or mathematical models that link ice cliffs and ponds to the dynamics. We will try to better include the more general qualitative relations from the literature here in the revised manuscript and better explain the issue of the lack of a quantitative model.

Lines 105-110. It would be helpful for the reader to include the equation for driving stress here. That way readers can connect to the fact that driving stress scales with ice thickness and surface slope. Is there a physical mechanism why cryokarst features might follow driving stress? Would be good to include that.

As mentioned in the main comments C2, we will add more model details here and add the driving stress equation.

Line 122. Just need to clarify what the CFL condition is here as this is the first time this acronym shows up in the text.

Agreed – we will add this.

**Section 3 Modelling results**

This section is rather difficult to follow and I am quickly overwhelmed by the number of simulations and how quickly the writing moves between them.

132-135. It could be beneficial to include a what you refer to as a 'baseline' case (with base debris concentration) so the reader has a single simulation to compare the others to. Reading below it is easy to get lost in all of the simulations. Maybe this baseline case could be bolded in the figures below?

Good point – we will revisit this and will more clearly identify and delineate the baseline case from what follows.

Figure 1. Nice figure. What part of the glacier is covered with debris? How does that relate to the ELA. Perhaps adding these would be helpful to bring the various components of the model together for the reader.

It seemed clear to us where the debris is here, since it is also included in the same figure. However, as the reviewer's comment speaks to the important point raised in the main comment C5, we will modify the glacier profiles in Fig. 1a and 1c to make clear that the debris cover starts at the ELA.

Line 157 need a hyphen between 'debris' and 'covered'

Yes – thanks, we will correct this!

Section 3.2 I think these are all important interesting simulations. This section would be improved though with a bit more synthesis. It is a bit difficult to follow because of the number of different experiments. Maybe more clear topic sentences clearly keying on what each experiment the paragraphs correspond to would help? Or sub-section titles for each experiment?

It might also be that the description moves between simulations using different englacial debris concentrations quickly. Perhaps it would be easier to follow if the descriptions of the experiments use one concentration case?

These are good points. We will better clarify, motivate, and differentiate the experiments here to make them more digestible to the reader.

Figure 2 is really a great future! 223-224. Might be good to have a citation here.

Agreed – we will add this.

Figure 4 is also really clear.

Thanks!

Table 4. The table looks very clean but maybe adding in text at the top the definition of each variable again would help the reader follow.

Agreed – we will add this.

Figure 6. The introduction of 'Bare ice %' is hard to wrap my head around since it seems to be a new way of describing the cryokarst features. Maybe just label it % of the surface composed of cryokarst features. Consider finding another way to represent the contribution of cryokarst that is more clear.

Good point – we will explain this term better in the text and consider an alternative wording.

Figure 7. You might consider moving this figure into the supplemental and just describing the effect of cryokarst on long term evolution in the text. Just so the reader does not feel overwhelmed.

Good point – we will consider moving this to the supplement and if not, then we will try to make the text associated with the figure clearer.

**4.1 Debris-covered glacier memory**

This section highlights some interesting findings. The section, though, would be improved by stating what past studies have concluded related to this topic and then emphasizing showing how your results/conclusions differ. This is especially relevant to interface a bit with past transient glacier model simulations. Do they show a similar effect that support your discussion here?

As agreed in the main comment C1 above, we will better situate and integrate our work in the context of previous literature on this topic.

299-302. This paragraph would benefit from a look at the past literature on the subject, as this point has been raised previously. Additionally Clark et al., 1994 et al. also discuss this effect.

We have found hints of this in the literature but nothing explicit. For example, the paper by Clark et al., 1994 discusses many well-known issues relating to the retarding effect of debris cover on glacier response but we could not find any mention of the idea that the length of a debris-covered glacier depends on the history of its cold phases. However, we will include this relevant reference in the introduction and discussion.

Line 347 -351. There are studies that do connect ice cliff occurrence to ice dynamics, including Benn et al., 2012; Kraaijenbrink et al., 2016, .

We already cite Benn et al., 2012 immediate above this but we will clarify this text and also consider citing Kraaijenbrink et al., 2016. However, it does not change the point we make here – that the onset of cryokarst features is quantitatively not well (or not very explicitly) linked to observations of glacier dynamics.

**Section 4.4 Steady state velocity–debris thickness relationship**

I think this is a very interesting section. I do think it would be improved if it interfaced with the previous literature on the topic. Especially emphasizing how this work has expanded on those previous insights.

As stated in the main comment C1, we agree with this and will add additional references

and context.

392-393. A & A 2018 also do a compilation of 8-10 glaciers that show that debris thickness patterns follow this same pattern. These observed profiles can also be referenced with the Mölg study as well.

Agreed – we will add this point.

395. "It is natural to ask to what extent the debris thickness profile depends on the ice flow model and the debris transport model used. That question can be answered for the steady state case without assuming anything about the ice flow and considering only conservation of mass."

It is unclear how the statement above relates to the rest of the paragraph. This seems like an interesting topic though.

This statement is directly connected to what follows, as we do not assume an ice flow model. We will adjust the text here in order to clarify this point.

Equation 11 is very similar to one derived by Anderson and Anderson (2018) who follow a similar approach. It seems appropriate to cite that you are following that line of logic or interface with that work here.

It does makes sense to cite A & A 2018 in a way that more clearly links this section with that study and we will do so. However, we start from a different perspective as we do not assume velocity is constant (as they do) but rather start with conservation of mass. We will clarify this difference here.

404. How is it possible that there is ice flow at the terminus that is not 0? The SIA is based solely on internal deformation which is requires that ice thickness is larger than 0 which is not the case at the terminus. Just a bit of clarification will help.

Again, we do not assume SIA in this section (and will clarify this in the text). We only assume conservation of mass. Also, even in our main study, our model does not necessarily involve zero ice flow at the terminus, as is discussed in the model development on lines 124-127. In fact, in steady state the ice flow speed at the terminus is always nonzero. In some retreat experiment the velocity can however go to zero.

409. There is an interesting discussion to be had between the insights from A &A 2018 (Fig. 9) and what is discussed in this paragraph, especially regarding the zone of englacial debris emergence as described there. How does this discussion mesh/build off of with what was discussed in A &A 2018?

Good point – we will add a further reference to A & A 2018, specifically linking this section to their Fig. 9.

**4.5 Model limitations**

421-425. This is a repeat from a point made above. My sense is that this only needs to be stated once.

Good point – we will remove one of the two references so that we make this point only once.

426 to 431. The authors should discuss the implications of the assumption of uniform englacial debris concentration further. From my view it seems more fair to say 'that the effect of a uniform englacial debris concentration should be explored further.' I mention this because there are a number of simplifications that go into this assumption.

I think its is a reasonable first order approach, but this means the entire ablation area will be covered with debris.

It needs to be added here that different ice flow paths will change the englacial debris concentration even with a uniform input of debris everywhere on the glacier. It is really impossible to have a glacier with a uniform englacial debris concentration because of the straining of ice and the inevitable variability of debris input (in space and time) to the glacier surface.

One additional point that should be discussed is how applying a uniform englacial debris concentration relates to headwall erosion rates. If the headwall erosion rate is constant in time then as a glacier gets bigger the englacial debris concentration by definition must become smaller.

This effect is not included in this model. By keeping the englacial debris concentration uniform and steady there must be a requisite increase in headwall erosion rate as the glacier grows in size. If the glacier doubles in size then the headwall erosion rate would need to also double. I think this is simply an underlying assumption of this approach that should be clear to the reader and if possible should be quantified and placed in the supplemental material.

This is an important point, as we note in the main comment C5 above. We agree with the reviewer and will clarify the implications of this assumption. We will also discuss how this affects the interpretation of our results.

427. missing period.

Thanks – we will add this.

434. It should be made explicitly clear what the differences are between the toe condition applied here and the one presented in A and A (2016) in the main text. Is it simply a modification of the approach presented in A and A 2016? Are they not also quantitatively similar? See the text regarding the toe parameterization in the Appendix below.

As stated in the main comment C2, we agree to state more clearly how our boundary condition differs from A & A 2016.

436-437. I could not find where this statement is discussed in the appendix. Is there a citation that notes this or is it a new observation? I am unaware of this effect.

We will clarify this point in the appendix.

437-438. The way this paragraph is written implies that A &A 2016's approach would not capture the effects of a stagnating tongue. Is this actually true? Looking at the other publications after A &A 2016 like Crump et al., 2017 and Anderson et al., 2018 the length change curves are similar to those presented here and based on the toe parameterizations the dynamics should be represented similarly to the work here.

We do not believe this paragraph implies anything about the limitations of the boundary condition used in A & A 2016. The context of this paragraph is on limitations of our model, not the model used in any other studies.

**Appendix A and the toe parameterization in general**

It is a substantial effort to develop a toe parameterization and any improvements on the exploration from A & A 2016 are welcome, important, and vital for the future development of debris-covered glacier models. It is also important that the method presented here also be reproducible. It would be good for the authors to describe the sub-grid interpolation scheme in detail. What shape/formula do you assume? What $H_*$ terms are viable?

465-475. It seems like there should also be some discussion of how this formulation relates to the original terminal condition described by A & A 2016. How are the approaches different?

A & A 2016 explore a range of possibilities for the terminal parameterization which the toe is drowned in debris because it cannot leave the glacier and also a case in which an ice cliff persists at the terminus and debris is effectively rapidly removed from the glacier. See Figure B1 and section 5.2 in A & A 2016. Ultimately, A & A 2016 use a scheme where debris is removed based on the bare ice melt rate which is basically the same as is implemented here. I think it would benefit the readership to have a more complete description of the differences between the two schemes and how different they actually are.

It seems that the parameterization presented here is a smart approach. Despite the way the text is written it seems the approach follows the A & A 2016 formulation closely and fits almost within the range of parameters explored there. The new approach presented here essentially sets no limit on the d_flux term from A & A 2016, and the formulation presented here would be close to the c =10 case in Figure B1 from A & A 2016 for debris removal. The main difference is that this approach keeps the ice cliff backwasting at the bare ice melt rate despite the removal of more debris than that backwasting of the ice cliff actually would allow.

The down side of the approach presented here is that the removal of debris from the glacier is not necessarily physically representative of the process of debris removal at the terminus of real debris- covered glaciers.

The A & A 2016 scheme honors that the removal of debris from the toe in the ice cliff case is determined by the backwasting rate, but this in turn leads to a greater grid scale dependence than the scheme presented here. From my view the benefit of either of these schemes depends on the decision to value either grid-scale dependence or the physical representativeness of debris removal from the toe.

Either way a more nuanced description of this toe scheme and how it relates to the work of A & A 2016 is needed to ensure the community can follow these methodological differences.

As we state in the main comment C2 above, we intend to better explain our toe boundary condition and to clarify the differences in our model boundary condition at that used in A & A 2016.

Figure B1. It seems like this figure should plot the mass balance curve with time, since the the cryokarst parameterization adjusts that directly. I would also like this figure with the SMB curves included in the main manuscript since the cryokarst parameterization is a central, new contribution of this work.

Agreed - we will add SMB plots in the main text and the appendix.

**General comments (Reviewer 2: Fabien Maussion)**

• At the end of the introduction, you write: "*to date no study has used a coupled ice flow-debris transport model to study in detail the transient response and characteristic response times of a debris-covered glacier. This study aims to fill this gap...*". "It has never been done before" is not a good motivation for a study, and I think that the paper would gain from clearly stated research questions. In particular, it would help to understand what motivated the model design and the design of the idealized experiments (why this bed profile, why this model design, etc.). Research questions will also help to place the study in the context of previous literature, and prepare the reader to understand what you are trying to achieve with this paper.

Agreed – as stated in the main comment C1 above, we will more clearly contextualize our study in light of what has been done before and this should better motivate our approach.

• The word "idealized" does not show up in the title, abstract, or introduction. I think it should be clearly stated much earlier (maybe not in the title, but at least in the abstract). "Numerical modeling" could be understood as "applied to real glaciers".

Good point! We will emphasize that these are idealized studies by stating this explicitly.

• This may be subjective, but I don't find any of the comparisons with Jóhannesson's response times informative or useful. Even without debris cover, you can find numerical response times of glaciers which are widely different than the an-alytical ones, since the e-folding times are highly dependent on parameters such as bed depressions, mass-balance (MB) gradients, etc. (see e.g. Zekollari et al. 2015 or Schuster, 2020 - unpublished thesis work).

While it is certainly true that one can generate widely different response times due to different geometry and/or climate forcing, here we have the same bed geometry (and very similar upstream surface geometry) and the same climate forcing. However, the response is quite different. Since the Jóhannesson time scale is often used to give a

general idea of how changing the geometry or forcing changes the response time for debris-free glaciers, we still feel that a comparison here is warranted. The purpose of the Jóhannesson time scale is actually to be able to compare glacier dynamics between different glaciers and it is a useful measure that characterizes the dynamic response. However, we can include a caveat that explains the limitations of this approach.

- Your code availability statement ("available upon request") is against this journal's data and open science policies: https://www.the-cryosphere.net/policies/data_policy.html. I strongly recommend to make your code available (under a clear license), which will increase the visibility and re-usability of your work.

We will make our code available when we submit a revised version of the manuscript, in accordance with TC's open science policy.

**Specific comments (Reviewer 2: Fabien Maussion)**

**Abstract** I'm not very familiar with the debris-covered glacier literature, but I had to search for "cryokarst" online

We will better motivate and explain our use of this term in the text.

**Abstract** add "idealized numerical simulations"

Agreed – we will add this.

**eq. (1)** consider using b instead of a for mass-balance (more common I believe)

We used $a$ instead of $b$ so as not to confuse with the bed $b(x)$ but we can state early in the manuscript that we are going against convention, just to clarify for the reader.

**L72** "for a given a bed elevation" - remove "a"

Agreed – we will fix this.

**L96** having read section 2.1.4 and the appendix, it's still not clear to me how you compute $H^*$ (and I don't want to check up on Anderson et al 2016). I notice later that $H^*$ is a constant and a model parameter: mention this earlier in the text.

Good point – we will clearly state that $H^*$ is a model parameter here and we will also better motivate how we choose this value.

**L100** specify which appendix.

Good point – we will fix this by identifying which appendix.

**Appendix A** despite of your valid attempts to show that this boundary condition may be found in the real-world, I still believ that the ice-free terminus condition is more a model necessity (trick) than a real-world feature. You don't have to change anything in the text

here, I just wanted to comment on that.

That is an interesting comment. While we admit that we do not know exactly what the best boundary condition is, we believe our formulation is consistent with a number of observations and importantly it is also grid-size independent. But we are happy to admit that there might be a better way to formulate the boundary condition here.

**Sect. 3.1 (steady states)** I really had to think twice about how you can reach steady state with such a model. I think that it would help to write more about it. E.g. by saying again that (i) steady state can be reached only because the MB doesn't go too close to 0 and (ii) that this is only possible by removing debris at the terminus and effectively capping the debris thickness to a reasonable value. You can refer to Fig. S1 in this section (or mention typical values of MB at the terminus in the model) to help understanding.

We will clarify here how a steady state can be reached.

**L155** to our knowledge, Jóhannesson et al, (1989) wrote:"The volume time-scale tau can be computed from the volume differences between two steady-state profiles scaled to the causal mass- balance change", but did not mention the e-folding volume response time (yet). Maybe refer to another paper as well: e.g. Oerlemans (1997) or Jóhannesson (1997)

Good point – our text implies that Jóhannesson's paper used *e*-folding time and we will adjust this.

**L185** volume-area scaling: since it might be unclear to some of your readers, add here that (in your model) area is directly proportional to length

Good point – we will add this.

**L190** is "stagnant" the correct term here? I was confused several times in the manuscript about this, because you seem to use "stagnant" for when the glacier length does not change. Personally, I understand "stagnant" as "ice that is not moving" (u = 0). You cannot have "stagnant" ice with your numerical model setup. I would argue for using "stable terminus" in place of "stagnant", or clearly state in the text what you mean with "stagnant". At the very end of the paper there is a sentence going in this direction ("*stagnation or more specifically the cessation in local dynamic replacement of ice*.").

**L277** "stagnated": same here. Is it the correct way to say that? Non-divergence is still happening with u constant and non-zero, i.e. moving ice.

We will clarify our use of the word 'stagnate', which is used in numerous other studies of debris-covered glaciers in the same way as we mean it. That is to say, stagnation here means that the glacier is no longer very dynamic. It is not necessary for the velocity to be zero in a zone of stagnation.

**Section 3.3** "white noise" traditionally, white noise climate should be applied on a year to year basis and the periods of cold and warm climates would occur "naturally", as a result of random sampling. I wonder how this would affect your results. Additionally, I wonder if an annually varying MB would still work with your debris cover formulation, since you don't deal with temporary ice/snow cover on debris as for now.

Agreed – we will change the wording here and we will not use 'white noise' anymore. We have run simulations with annually varying ELA as well and it works fine with our model but whether this is physically realistic is open to discussion.

**Fig. 5** while this figure carries well your main message, I think that it can be misinterpreted. In particular, the blue line in Fig. 5b gives the impression that the glacier will always grow, i.e. never reach a "steady state" (i.e. an average length around which it oscillates - albeit in a strange, debris covered way). What you could do here is continue the simulation for an additional 5k years (at least) and see what happens. It might have an interesting consequence: the "average length" of a debris covered glacier under a *random* climate might be longer than the length of the same glacier under *constant* forcing. I expect the average length to be some- where between the steady state lengths with the two ELAs (although it might even be longer than that, which would be very interesting to discuss further).

We attempted to raise these very points by showing these plots in Fig. 5. There are a number of further points that could be raised here and indeed, further studies one could do with our model to examine the volume and length response given different climate forcing. We will add some text here to clarify what this part of our study implies.

**Figure S1 :** write that "SS" stands for "steady state" in the legend.

Agreed – we will add this.

---

## Referee Report (RR1)

Thank you for taking the time to revise the manuscript. It is much improved and, from my view, close to being ready for publication. There are however some model descriptions that are not yet accurate and greatly hamper the ability of the reader to understand what the model actually does. Some comparisons to other work are also inaccurate and should be adjusted before the manuscript is ready for publication.

The first relates to the lack of clear explanation regarding the **toe parameterization** used in this study.

**Grid size dependence**
The toe parameterization presented here is in fact grid-size dependent but it is presented as if it is not in the main manuscript. Note that my comments reference the new updated version of the manuscript, not the marked up version. From Lines 129-131:

"We note that although the terminal boundary condition used here is similar to the one implemented in Anderson and Anderson (2016), since in both cases the glacier ends with a small debris-free terminal ice cliff, our choice of boundary condition has the subtle but important feature of being independent of grid size."

From line 570 describing the toe parameterization used in this paper:

"This boundary condition is found to be grid size independent in the sense that it results in steady state glacier extents that converge to a constant value as the grid size is decreased."

A strong contrast is drawn in the text between the toe parameterization approach of A & A (2016) and the one presented here (see lines 130-131 and 574-577). The grid size dependence is explained in A & A (2016):

"In our model, the debris thickness
h debris (x, t) represents a layer of equal thickness on any cell.
Debris thickens more slowly with a larger dx because the
debris volume advected into a cell is spread over a larger
area (due to the larger dx; dy=1; dy (in m)). There is there-
fore a timescale built into the thickening of debris in a cell
that is dependent on dx. Because ablation rates are sensi-
tive to debris-cover thickness, changing dx has an effect on
glacier evolution."

But there is virtually no difference between the model of A and A (2016) and those that follow and the new model presented here. Both models are grid size dependent and the effect will be reduced in both models as dx is reduced.  Please update the text so this false distinction between the models is removed. (see text in the Line-by-line comments below as well).

**Accurate explanations of the toe parameterization**
The way that the sub-grid ice cliff location is found is not yet explained in Appendix A. Line 564: "Since the critical ice thickness will generally fall between two grid points, a sub-grid interpolation is performed to determine the exact location of the ice cliff"

It seems like some sort of power function is shown in Figure A2 to estimate sub-grid the terminal location. Looking to the code though it looks like the x location of the ice cliff is determined by linear

interpolation between the numerical terminus of the glacier and the next cell. This does not match what is shown in Figure A2. Please update Figure A2 to show what actually happens in the code and the text to explain the actual means of interpolation used in the code.

Furthermore it is not clearly stated that the melt in the last cell is based on melt rates that are an average of the glaciated and non-glaciated parts of the cell (see equation A1). This is a non-physical representation of the melt rate at the toe of a DCG. This is not necessarily a problem, it is just not clearly stated. Tying Figure A2 to equation A1 is also needed as right now they do not reflect what actually happens in the code. This includes defining $x_*$, $x_i$ and $x_{i+1}$ in Figure A2. Without this explanation the reader cannot reproduce the toe method presented in the manuscript.

Line 563 "All the surface debris transported past the icecliff location $x*$ slides down the cliff and out of the system."

The model real does not represent debris transport at all after cell $x_i$ and there is no ice cliff backwasting that removes debris from the glacier surface. In the actual model any debris that flows beyond cell $x_i$ is removed from the glacier because it is set to zero in the code. Please update this description as it does not accurately represent how debris is actually removed from the modeled glacier.

There are really only two ways debris can leave a glacier 1) by ice cliff backwasting and 2) by gravitational mass wasting (A&A 2016). These processes are not represented in the model presented here.

**Constant englacial debris concentration and DCG memory**

One important conclusion presented in the abstract and conclusion (that cold periods will disproportionately increase DCG length/size causing a form of hysteresis) is consistent with the effects of the assumed constant, uniform englacial debris concentration. As a glacier grows due to colder temps, more debris will be present in the glacier across entire ablation zone than should be for a steady headwall erosion rate. This will tend to increase the glacier size due to a non-physical increase in debris on the glacier surface. Because this hysteresis is consistent with the effects of constant englacial debris concentration in a glacier that is changing size I am not convinced that this hysteresis is a real effect for DCGs. The non-physical nature of the toe parameterization may also contribute to this but it is hard to evaluate this without taking into account the conservation of debris mass in the model presented here.

Here is what the model of A &A (2016) does when working through a climate cycle allowing for englacial debris advection (figure 20 from Anderson et al., 2018):

[Figure]

There is very little change in maximum glacier length despite cold and warm cycles. Is the amplification seen in this model is in fact dependent on the constant englacial debris assumption. I note the caveat that this model is meant to represent a small DCG transitioning into a completely DCG (rock glacier) but the enhancement effect is absent outside of the first cycle.

Compared to figure 6 from this study:

[Figure]

**Line-by-line comments**

Line 56 "has also studied the relationship between debris-covered glaciers and rock glaciers."

Please add 'transient' between 'the' and 'relationship' otherwise the way this is written makes it seem like the model is only being used in a steady state sense from the text immediately before.

Line 109: "Østrem curve data representative of a medium-sized Alpine glacier"

Consider re-vising this justificatoin as it is unclear what the size of a glacier has to do with the shape of Østrem's curve.

Line 126: "In addition, we accounted for the fact that the terminal ice velocity in SIA goes to zero, which is not physically realistic, by taking an averaged velocity over several ice thicknesses near the terminus when computing the debris transport."

How does the ice not go to zero velocity at the toe on a real glacier? If ice thickness is 0 velocity must be zero. How physically does ice flow remove debris from a glacier? This is not the correct process for the removal of debris at the toe, which can only leave by ice cliff retreat or by geomorphic trundling down the glacier slope.

Line 130-132. "We note that although the terminal boundary condition used here is similar to the one implemented in Anderson and Anderson (2016), since in both cases the glacier ends with a small debris-free terminal ice cliff, our choice of boundary condition has the subtle but important feature of being independent of grid size."

I see in the appendix that the terminus formulation presented in this manuscript is in fact grid size dependent.

Line 132-133: "Further details of the boundary condition, including the interpolation scheme between grid points and convergence tests for different grid sizes, is found in Appendix A."

I did not find a reproducible explanation of the interpolation scheme and how it differs from A & A, 2016 nor the convergence tests for different grid sizes in Appendix A. Please update accordingly.

Figure 1. Nice addition!

Line 174. Helpful explanation of what is to come.

Line 354. "We showed that the memory of a debris-covered glacier is selective, exhibiting an effective hysteresis, with periods of relatively cold climate having a sustained effect on the volume and in particular on the length."

While this is a clear result from the modeling and model set up (Fig 6), I am concerned about the effect of an assumed constant englacial debris concentration through time on this conclusion.

Perhaps one way around this issue is to state that the effect you see with glacier size enhancement at cooler temps and glacier shrinking at warmer temps is consistent with the englacial debris concentration changing in time but that it appears to be independent. I am not sure what simulations can be done with this model to show that this hysteresis is in fact not model assumption dependent.

Line 357. "Previous numerical simulations of the transient response of debris-covered glaciers focused only on the effects of sudden debris input in the form of an avalanche (Vacco et al., 2010; Menounos et al., 2013). Such a one-time debris input leads to an advance in glacier extent and foreshadows the results of our study, where a constant debris source and changing climate forcing gives rise to a more complex response."

This is statement is not accurate and omits a number of other studies of the transient modeling of DCGs in response to climate change.

Other studies have modeled the transient response of DCGs to climate change with a steady input of debris. Not only a one time avalanche of debris. Please the Rowan et al. (2015) model of Khumbu glacier responding to a warming climate. See Anderson et al., 2018 (Glaciation of alpine valleys: The glacier-debris-covered glaciers continuum) and Crump et al., 2017 for examples of modeled glaciers that contradict this above statement.

Studies that show transient response of debris-covered glaciers to climate change:

Rowan et al. (2015). The whole paper focuses on transient responses to climate change with a steady debris deposition rate.

*Anderson et al., 2018*: See figures 18 and 19 (for a single climate cycle) , and figure 20 for the transient change in a debris-covered glacier over 7 climate cycles).

*Crump et al., 2017*: See Figures 6 and 7 show a debris covered glacier responding to step changes in ELA over a 4000 year period.

The statements in the text here need to be adjusted to reflect the contributions of these previous efforts.

Line 363. "This also means that for debris covered glaciers, no unique glacier length exists for a given climate, but rather that the length of debris covered glaciers is determined by the history of repeated cold phases. Furthermore, debris-covered glaciers under random climate forcing are
expected to have a longer average length than the steady state length corresponding to the equivalent constant climate forcing."

See comments above.

Line 439. Maybe should be up to 12% here based on Anderson et al., (2021).

"Kennicott Glacier exhibits the highest fractional area of ice
cliffs (11.7 %) documented to date."

Line 476. "Equation (14) is similar to an equation derived by Anderson and Anderson (2018) but with the important difference that we have allowed variable velocity in this derivation, rather than assuming a constant velocity along the entire glacier."

Which equation are you referring to here from A & A (2018)?

There are two models presented and discussed in A & A (2018). And variable velocities are explicitly discussed and modeled in that paper. So I am confused as to what you are referring to. Please clarify in the text as it creates a false sense that spatially variable velocities were not considered in A & A (2018).

**Appendix A**

See notes in the main comments above.

Line 575. "A similar boundary condition was used in Anderson and Anderson (2016), where a debris-free region was also employed at the terminus. In their approach, the size of the terminal ice cliff depends directly on the grid spacing but the debris flux can be varied. In contrast, our terminal ice cliff is roughly independent of grid size, since it depends on the critical ice thickness H $*$ , but the debris flux is fixed by the local ice velocity."

This is not correct and should be adjusted. The ice cliff height in A & A 2016 does not vary depending on grid size and the ice cliff height is fixed. There is no debris-free region on the modeled glacier. The grid size dependence is explained in A & A (2016):

"In our model, the debris thickness
h debris (x, t) represents a layer of equal thickness on any cell.
Debris thickens more slowly with a larger dx because the
debris volume advected into a cell is spread over a larger
area (due to the larger dx; dy=1; dy (in m)). There is there-
fore a timescale built into the thickening of debris in a cell
that is dependent on dx. Because ablation rates are sensi-
tive to debris-cover thickness, changing dx has an effect on
glacier evolution."

This effect is the same in this model as well. As dx is decreased the effect of dx on glacier evolution is reduced. It seems that the model of A &A (2016) and the  model presented here have the same issue, namely that there is grid size dependence in the length that converges as *dx* is reduced. Please update the text to accurately represent the similar issues for the models as right now the text states that the model presented here is grid size independent when it is not actually.

---

## Referee Report (RR2)

Thank you for your thorough replies. I appreciate you taking the time to clarify the issues raised.

There are only couple of small items that should be clarified/revised:

Line 489-490.
"Equation (14) is similar to equation:::: (27) derived by Anderson and Anderson (2018) but with the important difference that we have allowed variable velocity in this derivation, rather than assuming a constant velocity along the entire glacier."

Equation (27) in A and A, 2018 does not assume uniform velocity for any part of the glacier, rather it only assumes that englacial debris emergence is negligible. It is derived directly from the continuity equation for debris and is independent from the analytical model presented in that paper (A and A, 2018). Please revise the text above to reflect the fact that variable surface velocities are not assumed in the derivation of equation (27) from A and A, 2018.

The text from A and A, 2018 is below for completeness:

**4.3. Concave up debris patterns: velocity controlled**

Debris emergence is negligible if there is no debris within the glacier or if sub-debris melt rates are small. In such cases Eq. (9) reduces to

$$\frac{\partial h_{debris}}{\partial t} = \frac{\partial (h_{debris} U_s)}{\partial x} \tag{26}$$

Solving for $h_{debris}$ by taking the integral of Eq. (10) with respect to $x$ leads to

$$h_{debris}(x) = \frac{q_e}{U_s} \tag{27}$$

where $q_e$ is the surface parallel flux of debris at the down-glacier end of the emergence zone (where the emergence rate declines to zero; see Glazyrin, 1975). Here $q_e$ is constant with respect to $x$ and is the integral of emergence rate. Hence

$$q_e = \int_{x_e}^{x_{e_{end}}} \frac{Cb'}{(1-\phi)\rho_r} dx \tag{28}$$

And equation 9 from A and A, 2018:

In the ablation zone, the rate of change of surface debris thickness in one-dimension may be written

$$\frac{\partial h_{debris}}{\partial t} = \frac{Cb'}{(1-\phi)\rho_r} - \frac{\partial(U_s h_{debris})}{\partial x} \qquad (9)$$

where $b'$ is the sub-debris ice melt rate, $U_s$ is the down-glacier surface speed, and $\rho_r$ is the density of rocks comprising the debris cover (e.g., Nakawo et al., 1986; Naito et al., 2000). The first term on the right-hand side is a local debris emergence rate $\varepsilon_{debris}$ (in units of m y$^{-1}$), and the second represents the thickening or thinning rate of debris caused by down-valley gradients in the surface debris discharge $U_s h_{debris}$. For the analytical model, we assume a uniform $U_s$ field in the debris-covered portion of the glacier so the right-hand term in Eq. (9) is neglected.

Line 602.

"A second difference is the fact that the size of the terminal ice cliff — and hence the size of the region over which the terminal debris flux condition is applied — depends directly on the grid spacing, resulting in a smaller ice cliff as the grid size is reduced. The implications of this for model convergence are not clear."

Ice cliffs are defined quite differently in the two papers/terminal parameterizations.

The terminal wedge parameterization in A and A, 2016 prevents the issue outlined by the authors above for the grid sizes explored there (100 and 200m). The ice cliff height is set to 10 m, is vertical, and is applied within the terminal wedge.

Here is the ice cliff parameterization as done in A and A, 2016:

[Figure]

The ice cliff height does not change in A and A, 2016 because it is defined as a parameter in the code and the ice cliff height is always much smaller than the ice thickness at the start of the terminal wedge. The code from A and A, 2016 is not yet available (it should be and I would like to make it so) so this is not clear to the authors here.

I would simply appreciate if the text quoted above was at least edited to reflect that the ice cliff is defined differently between the papers and as it is defined in A and A, 2016 the ice cliff height is not grid size dependent.

It is accurate though to say that the height of the start the terminal wedge is grid sized dependent. But because the terminal wedge is debris covered in A and A, 2016 it does not introduce ice cliff height dependence issues as described by the authors.

Best wishes for finalizing the manuscript!

Leif

---

## Editor Decision (ED1)

Dear James Ferguson and Andreas Vieli,

Thanks a lot for your fast response to the last reviewer comments and for updating the manuscript accordingly.

It is my pleasure to accept your manuscript for publication in TC. I had a last careful read, and really enjoyed the quality of the material presented, the very smooth storytelling aspect and the nice and effective visualizations. I have formulated a list of final comments and suggestions that I hope will help to (further) clarify a few elements. It would be great if you could consider these suggestions and incorporate them when submitting the final version of your manuscript:

- l. 19-20: "This debris becomes entrained in the ice...": is this always the case? Sometimes debris can also directly fall on the surface in the glacier's ablation area, no? Possibly consider rephrasing to "This typically becomes..." and mentioning somewhere the possible direct supply of debris on the glacier surface in the ablation area
- l.31-32: mass loss rates similar for some debris-covered glaciers vs. ice-free glaciers. Long list of references here. Maybe good to also have some recent large-scale works that clearly show this from geodetic remote sensing observations? (Shean et al., 2020; Hugonnet et al., 2021) + potentially also add these for your statement at l.387.
- l.41-42: "...provides the only observable record...": are there really no others? Probably not as detailed, but I imagine that similar observations can be derived for other glaciers (e.g. by comparing old maps/paintings/...etc). Consider rewording to e.g. "...provides, to our knowledge, the most detailed observable..."
- l.84: "the thickness evolution": just to avoid any possible confusion, e.g. between ice and debris thickness, would suggest explicitly mentioning "the ice thickness evolution" here
- l.87 + Table 2: why was a value of $1 \times 10^{-24}$ Pa s$^{-1}$ chosen here? In the literature there is quite a large spread for this value for temperate glaciers, and an often used reference value is the one by Cuffey and Paterson (2010) of $2.4 \times 10^{-24}$ Pa s$^{-1}$. Would your results and the response time be very different under such a value? Not suggesting that elaborate analyses should be done at this stage, but would be good to probably mention this somewhere in the discussion, potentially complementing this with a (rough) estimate about how your response times may be affected with this quite different value for the rate factor
- l. 102-103: "...close the the ELA and beyond". Not entirely clear what "beyond" is here. I guess "above" the ELA? Possibly reword to avoid any confusion
- eq. 7: I was a bit puzzled with the definition related to "smaller than" and "larger than x*": intuitively I would expect values "smaller than" to be somewhere at the glacier front, but this is the other way round here. When seeing your figures, I realize that this is because the highest glacier parts correspond to a low x-value. Maybe good to mention here how the x-values are defined, and explicitly state that the upper part of the equation relates to the glacier front and the lower part to the higher glacier parts? Related comment in l.337: where you describe the glacier front as the "last 200 m", while I would intuitively describe these as the "first/frontal 200 m" (+ similar remark for caption of figure 10 + l.342)

- l.131: "several ice thicknesses": when reading this, I was wondering how many ice thickness this would be and if more information could be provided. Later in the manuscript (l.167) I was happy to read that this information is shared. Maybe good to group this information in order to avoid redundancy?
- l.159: "realistic values taken to be...": can you give an indication based on what these values are thought to be realistic? (e.g. other modelling studies, observations,...)
- l.163-165: CFL. Could you explain a bit more extensively (e.g. one additional sentence) on what the CFL criterion is based? This is mostly trivial for ice flow modellers, but may be difficult to grasp for others that are not familiar with this concept. Moreover, I found the last phrase, which is most likely entirely correct from a theoretical perspective, difficult to understand: "...which is necessary to... domain of dependence". So potentially consider replacing this last sentence with a more general explanation about the CFL criterion (+potentially add the original reference also: Courant et al., 1928)
- l.190: why where ELA's of 3000 and 3100 m chosen? Seems rather arbitrary. Maybe because this typically corresponds to ELA of a well-known glacier (Zmutt) and that the changes correspond to typical changes between LIA ELA and present-day LIA? I am just guessing, so would probably be good to have a hint about how these were chosen, especially given their central importance in your work (all results are directly related to this choice)
- l.200: "almost identical": I was wondering why they are not entirely the same? Given that you rely on the SIA, and not a solution that accounts for longitudinal stresses, , the local geometry and ice flow do not 'see' what happens in the lower glacier parts (which is where the forcing at the surface is different, in the upper parts the forcing is exactly the same, right?). May again be related to my misunderstanding but would be nice to have a hint about why they are not exactly the same.
- l.251: "...because dynamic replacement of ice is close to zero": this means that the local thinning is then equal to the local mass balance if I understand correctly? If so, may be worth mentioning, as this is a quite 'intuitive' / 'easy to interpret' finding (potentially also referring to eq. 1). Same in l. 351: maybe also mention this here? (e.g. "...the total amount of thinning, almost equal to the local SMB, is large enough...")
- Fig. 3+7: why is there a sudden drop in the volume before reaching the lowest value (e.g. for blue curve in figure 3b around year 600)? What process causes an almost instantaneous large ice loss? Or is this a model artefact (e.g. related to space/time discretization)? Would be great if a hint could be given (or maybe this is the case and I missed this?)
- Section 3.3: no comment, but rather an appreciation. I really found this section to be very nicely presented and an important finding. Personally, my 'highlight' of your findings!
- l.287-288: why is a range chosen from 0 to 20%? Are there observations that these values are always below 20%? If so, which ones? Or indications from other modelling work? Now this seems to be a rather arbitrary choice, which I guess it is not
- l. 302: "effects" → "affects"?
- l.353: "...amount of thickening at the terminus": is there also a readvance then? (even if small) If so, mention this?
- l.379: "...and stagnating tongues" → "and have stagnating tongues"?

- l.397: "ability of the terminus to retreat in response to several successive warm periods (several centuries)": maybe mention the figure where this can be seen
- l.412-413: "stagnation in dynamics": you are referring to "ice dynamics" here (as opposed to e.g. debris thickness dynamics) right? May be good to explicitly refer to "ice" here
- l.417: "since they are sub-grid scale" → "since they are occurring at the sub-grid scale"?
- l.418: order studies according to publication year
- l.423-426: found this sentence quite long and difficult to read. Consider splitting up in two shorter sentences?
- eq.11 (l.440): the definition of the ice thickness to be used in this equation is generally also a topic of discussion/debate (should the mean/median/maximum be taken). This will also influence the response times you obtain. Maybe worth mentioning this.
- l.506: "near the ELA": but this is still above the ELA, right? Maybe mention this more explicitly by changing this to "slightly/just above the ELA". And what about debris that directly falls on the ablation area surface (see also very first comment)
- l.523: "model with the capability to track englacial debris transport": maybe explicitly mention models which do this (Rowan et al., 2015; Wirbel et al., 2018)
- l.542: "the model is however much more responsive": compared to length, right? At first, when reading this I thought that this was compared to the observations. Maybe mention this explicitly: "..much more responsive compared to length changes and can..."

Many thanks for choosing TC to disseminate your work. I am convinced that this work will be a very valuable contribution to our community!

Kind regards,
Harry

**References**

Courant, R., Friedrichs, K. and Lewy, H.: Über die partiellen Differenzengleichungen der mathematischen Physik, Math. Ann., 100(1), 32–74, 1928.

Cuffey, K. M. and Paterson, W. S. B.: The physics of glaciers, Butterworth-Heinemann, Oxford., 2010.

Hugonnet, R., McNabb, R., Berthier, E., Menounos, B., Nuth, C., Girod, L., Farinotti, D., Huss, M., Dussaillant, I., Brun, F. and Kääb, A.: Accelerated global glacier mass loss in the early twenty-first century, Nature, 592, 726–731, doi:10.1038/s41586-021-03436-z, 2021.

Rowan, A. V., Egholm, D. L., Quincey, D. J. and Glasser, N. F.: Modelling the feedbacks between mass balance, ice flow and debris transport to predict the response to climate change of debris-covered glaciers in the Himalaya, Earth Planet. Sci. Lett., 430, 427–438, doi:10.1016/j.epsl.2015.09.004, 2015.

Shean, D. E., Bhushan, S., Montesano, P., Rounce, D. R., Arendt, A. and Osmanoglu, B.: A

Systematic, Regional Assessment of High Mountain Asia Glacier Mass Balance, Front. Earth Sci., 7(February), doi:10.3389/feart.2019.00363, 2020.

Wirbel, A., Jarosch, A. H. and Nicholson, L.: Modelling debris transport within glaciers by advection in a full-Stokes ice flow model, Cryosph., 12(1), 189–204, doi:10.5194/tc-12-189-2018, 2018.

---

## Author Response (AR2)

**Response letter to review for "Modelling steady states and the transient response of debris-covered glaciers" by J.Ferguson and A.Vieli**

Dear Dr. Zekollari,

Thank you for continuing to handle the review process for our manuscript. We have addressed all the reviewer's comments below.

Best wishes,
James Ferguson and Andreas Vieli

**1 Summary**

We would like to thank the reviewer for a further helpful review. In order to address the valuable suggestions and proposed improvements, we have substantially rewritten portions of the manuscript, included additional convergence testing results, completely redrawn the schematic figure and included an additional supplementary figure. In what follows, we show first the reviewer's comments in black and then our response in blue. At the end of this document, we provide a marked-up version of the manuscript which shows the changes that were made between the original and the newly revised.

**2 Major comments**

Thank you for taking the time to revise the manuscript. It is much improved and, from my view, close to being ready for publication. There are however some model descriptions that are not yet accurate and greatly hamper the ability of the reader to understand what the model actually does. Some comparisons to other work are also inaccurate and should be adjusted before the manuscript is ready for publication.

The first relates to the lack of clear explanation regarding the **toe parameterization** used in this study.

**Grid size dependence**
The toe parameterization presented here is in fact grid-size dependent but it is presented as if it is not in the main manuscript. Note that my comments reference the new updated version of the manuscript, not the marked up version. From Lines 129-131:

"We note that although the terminal boundary condition used here is similar to the one implemented in Anderson and Anderson (2016), since in both cases the glacier ends with a small debris-free terminal ice cliff, our choice of boundary condition has the subtle but important feature of being independent of grid size."

From line 570 describing the toe parameterization used in this paper:

"This boundary condition is found to be grid size independent in the sense that it results in steady state glacier extents that converge to a constant value as the grid size is decreased."

A strong contrast is drawn in the text between the toe parameterization approach of A & A (2016) and the

one presented here (see lines 130-131 and 574-577). The grid size dependence is explained in A & A (2016):

"In our model, the debris thickness h debris (x, t) represents a layer of equal thickness on any cell. Debris thickens more slowly with a larger dx because the debris volume advected into a cell is spread over a larger area (due to the larger dx; dy=1; dy (in m)). There is there- fore a timescale built into the thickening of debris in a cell that is dependent on dx. Because ablation rates are sensi- tive to debris-cover thickness, changing dx has an effect on glacier evolution."

But there is virtually no difference between the model of A and A (2016) and those that follow and the new model presented here. Both models are grid size dependent and the effect will be reduced in both models as dx is reduced. Please update the text so this false distinction between the models is removed. (see text in the Line-by-line comments below as well).

We apologize for the inconsistency and potential confusion with regard to the grid size dependency in the text. We addressed this issue by rewriting parts of both the Methods section and Appendix A, clarified the grid size dependency issue and also added a table in Appendix A and a new supplementary figure to demonstrate convergence. While we agree with the reviewer that strictly speaking our method is grid size dependent, we now more clearly state this and demonstrate convergence of the results for decreasing grid size. Note that for the grid size of 25 m used in the modelling presented in the paper, the dependency is weak (further see line-by-line comments for details).

We further clarified our method of the toe parametrizations (section 2.1.4, Appendix A and revised the schematic Figure A2 in Appendix A) and clarified potential differences to the approach by Anderson and Anderson (2016) (A and A). However, we would like to note here that in our manuscript we do not aim to focus on a model-model comparison, nor do we try to judge which model or toe parametrization is better. But rather we try to explain our approach, explain why we did it in this way, and highlight potential differences to the model of A and A. These differences are not always that easy to assess as we do not have access to the code of the A and A model.

**Accurate explanations of the toe parameterization**
The way that the sub-grid ice cliff location is found is not yet explained in Appendix A. Line 564: "Since the critical ice thickness will generally fall between two grid points, a sub-grid interpolation is performed to determine the exact location of the ice cliff."

It seems like some sort of power function is shown in Figure A2 to estimate sub-grid the terminal location. Looking to the code though it looks like the x location of the ice cliff is determined by linear interpolation between the numerical terminus of the glacier and the next cell. This does not match what is shown in Figure A2. Please update Figure A2 to show what actually happens in the code and the text to explain the actual means of interpolation used in the code.

It seems that our explanation may not have been clear enough and therefore we added further details on this. The ice cliff location is actually not found by interpolation with the glacier terminus at all, since the ice cliff occurs at a critical ice thickness. Rather, as we state now more clearly in Appendix A – and as is shown in the accompanying code – this sub-grid location is found by linear interpolation between surface grid points. We have rewritten the text in Appendix A to clarify this point and revised the related schematic figure A2 in the Appendix. See the line-by-line comments for more details.

Furthermore it is not clearly stated that the melt in the last cell is based on melt rates that are an average of the glaciated and non-glaciated parts of the cell (see equation A1). This is a non-physical representation of the melt rate at the toe of a DCG. This is not necessarily a problem, it is just not clearly stated. Tying Figure A2 to equation A1 is also needed as right now they do not reflect what actually happens in the code. This includes defining x*, $x_i$ and $x_{i+1}$ in Figure A2. Without this explanation the reader cannot reproduce the toe method

presented in the manuscript.

We need to clarify here, that we do not use an average of sub-grid melt rate but a weighted average. This is a crucial distinction because it allows our method to capture sub-grid effects at the transition from debris covered to terminal ice cliff more accurately. This weighting is already correctly represented by equation A1, where the weighting given to the melt rate components is proportional to their respective grid cell length fractions. In order to make this more transparent, we have adjusted the text and added more detail to Figure A2.

Furthermore, we believe this is a reasonably accurate representation of the physical processes that govern debris transport at the terminus, as is represented in the two real world examples depicted in Figure A1, where the terminus has a debris-free ice cliff. The reader has not only the equations which define the boundary condition but also the freely available source code, which will allow an accurate reproduction of the methods used here.

Line 563 "All the surface debris transported past the icecliff location x* slides down the cliff and out of the system."

The model real does not represent debris transport at all after cell $x_i$ and there is no ice cliff backwasting that removes debris from the glacier surface. In the actual model any debris that flows beyond cell $x_i$ is removed from the glacier because it is set to zero in the code. Please update this description as it does not accurately represent how debris is actually removed from the modeled glacier.

There are really only two ways debris can leave a glacier 1) by ice cliff backwasting and 2) by gravitational mass wasting (A&A 2016). These processes are not represented in the model presented here.

Again, it seems our explanation of the boundary condition was not clear enough and the reviewer seems to have misunderstood how our boundary condition works, since in fact in our model, debris leaves the system by both of these processes. Terminus backwasting occurs during a retreat, when the glacier cannot supply enough mass flux to keep the glacier at its current extent. This occurs in Figure 5b between $\Delta t = 0.8$ and $\Delta t = 1$ and is clearly seen in the accompanying animation. Gravitational mass wasting implicitly occurs when the debris falls off the glacier at a terminal cliff due to the effects of gravity (by removing all debris beyond the position x* at the critical thickness H*). This is the principal way debris leaves the glacier in our model and this is why we do not transport debris past the location of the ice cliff $x^*$, since it is assumed to fall off the glacier at a timescale that makes it irrelevant for the surface mass balance further downglacier. We added a few sentences in Appendix A to improve the general explanation of our toe parametrzation and how the debris leaves and further added some more details of how it is implemented in the code (see detailed line-by-line comments).

**Constant englacial debris concentration and DCG memory**

One important conclusion presented in the abstract and conclusion (that cold periods will disproportionately increase DCG length/size causing a form of hysteresis) is consistent with the effects of the assumed constant, uniform englacial debris concentration. As a glacier grows due to colder temps, more debris will be present in the glacier across entire ablation zone than should be for a steady headwall erosion rate. This will tend to increase the glacier size due to a non-physical increase in debris on the glacier surface. Because this hysteresis is consistent with the effects of constant englacial debris concentration in a glacier that is changing size I am not convinced that this hysteresis is a real effect for DCGs. The non-physical nature of the toe parameterization may also contribute to this but it is hard to evaluate this without taking into account the conservation of debris mass in the model presented here.

Here is what the model of A &A (2016) does when working through a climate cycle allowing for englacial debris advection (figure 20 from Anderson et al., 2018):

[Figure]

There is very little change in maximum glacier length despite cold and warm cycles. Is the amplification seen in this model is in fact dependent on the constant englacial debris assumption. I note the caveat that this model is meant to represent a small DCG transitioning into a completely DCG (rock glacier) but the enhancement effect is absent outside of the first cycle.

Compared to figure 6 from this study:

[Figure]

The reviewer suggests here that the tendency to stagnate and advance (see abstract) under a fluctuating climate forcing is due to the non-constant input of debris in our model. Although we agree with the reviewer that this non-constant debris input will have some effect and provides some limitations, we are confident that it does not affect our main conclusion of asymmetric length response under a fluctuation climate on century time scales. The evidence from both our paper and from additional experiments we report here does not agree with the reviewer's suggestion, which we explain in the following and also further clarify in the manuscript.

(i) We stress that our simple step experiments clearly demonstrate an asymmetry in length response (less so for volume response), meaning for a step-warming the length retreat is strongly delayed (stagnated terminus for several decades/a century). This is simply an effect of terminus stagnation due to the debris insulation and occurs despite a reduced debris input. In contrast, for a cooling the advance occurs almost immediately. This means for climate perturbation over century time scales the glacier terminus struggles to retreat (in a warming phase) but may be able to advance (for a cooling phase). Note that for longer warming periods (or several successive century long warm periods; see random climate experiment Fig. 9) the terminus has enough time to completely collapse and fully retreat (see for example after 1200 years and 4800 years in Fig 9c). We added some sentences in the

discussion (in model limitations 4.6) to clarify these points, for specific text see below and tracked changes in the modified manuscript.

(ii) We do not think the modelling experiment of Anderson and Anderson (2018) that the reviewer refers to is a valid comparison, nor does it truly disagree with our modelling results. We point out that the transient response of any model will be highly dependent upon the amplitude and frequency of the forcing function. The example from Anderson and Anderson (2018) mentioned by the reviewer uses a sine forcing function with an amplitude change of nearly four times larger than our random climate experiment ($\Delta T = 3°$ versus $\Delta T = 0.8°$) and a period of five times larger (1000 years versus 100 years at each ELA, which is roughly equivalent to a 200 year period). As explained above in (i), for much longer forcing periods of several centuries/millenia (as in A and A 2018) we also would get retreat (see for example after 1200 years and 4800 years in Fig 9c), even more so for much higher forcing amplitudes. Further, in our case we focus on debris covered glaciers that are very extensively debris covered (up to the ELA), unlike in the example shown by the reviewer. We added some clarification in the abstract and discussion (sections 4.2 and 4.6) on the dependency of the forcing time scale (tendency of advance on century timescales, but retreat reset on millenia time scales), for specific text see below and tracked changes in the modified manuscript.

(iii) We note that the change in debris input is not all that large and that the time it would take to significantly influence the glacier is much larger than the timescale of the forcing. For example, in the case of random climate experiment for $c = 0.25\%$, the volume ranges between $+16\%$ and $-10\%$ of the mean ELA volume and the mean velocity of roughly 10 m/yr means a total glacier transit time of roughly 1000 yr. While there may be a small anomalous effect resulting from the variable debris input, it would likely be negligible given that timescale of the forcing is 100 yr. We already note this in the discussion in the first paragraph of model limitations (4.6).

(iv) We further would like to stress again that the asymmetry in retreat and advance response and the tendency to stagnate and advance in length under a fluctuating climate seems consistent with the observed extensive and stagnated debris covered tongues in today's warming climate (which we already mention in the text).

(v) Importantly, we have performed a similar experiment using a more sophisticated numerical model that resolves the englacial debris transport and in which we can enforce constant total debris input under a changing climate forcing. While this new model is part of another study and hence outside the scope of the current manuscript, it shows similar results under the same random noise forcing. The following figure below shows in panel (a) the same random climate forcing as Fig. 6 and 9 from our manuscript (note we used the same magnitude of ELA variations on the same timescale but due to the differences in the model setup the mean ELA is slightly different). The time series of the volume, in panel (b), and the length, in panel (c), are plotted as solid lines, whereas the dashed lines are representing the corresponding mean ELA values of volume and length. In agreement with our results from the simpler model in our manuscript, we see that (i) the volume response lags the climate forcing slightly but that the fluctuations stay close to the expected (i.e. mean ELA) volume; and (ii) the length has again the tendency of stagnation or advance and gradually increases throughout the time series so that at $t = 5000$ yr, the glacier extent is roughly 1 km greater than the mean ELA length. Therefore even for a constant debris input, our conclusion of tendency of advance or length stagnation for a fluctutating climate on century time scale is confirmed.

We address the reviewers concerns and points made above by adding in the text a few more explanations for better demonstrating the robustness of our conclusions of tendency of advance/length stagnation for a fluctuating climate (despite our limitation of total debris input):

In the abstract we clarified: "fluctuating climate at century time scales"

In section 4.2: "The glacier termini seem to struggle to retreat in warmer periods even if they are sustained

[Figure]

Figure 1: Additional modelling experiment with a model that includes the tracking of englacial debris transport and ensures the same total debris input in the case of a variable climate. It shows well that even for constant debris input the tendency for advance or stagnation in glacier length (panel c) on century scale climate fluctuations (panel a; same forcing as in paper) persists, whereas for the volume (panel b) this effect is strongly diminished.

over a century, and hence debris covered glaciers have the tendency to either advance or stagnate in a century-scale fluctuating climate (Fig. 6)". ... and further down ... "This is indicated in the random climate experiments by the ability of the terminus to retreat in response to several successive warm periods (several centuries)."

In section 4.6: "Hence, our modelled tendency of advance or stagnation in a fluctuating climate is essentially a result of the asymmetric length response. For warm periods at time scales longer than the terminus retreat delay (e.g. several centuries) the random climate forcing experiments demonstrate the ability of the terminus to retreat substantially and to reset this memory to cold periods."

We further clarify earlier in the Methods (section 2.1.2) that our model is focussed/appropriate for the case of heavily debris-covered glaciers (with deposition down to the ELA or debris up to ELA, e.g. heavily debris-covered glaciers in Himalaya):

"Further, the assumption of uniform debris concentration within the ice means that debris will be present over the entire ablation area and hence our model representative of extensively debris covered glaciers with debris deposition close to the ELA or beyond (e.g. Himalaya)."

**3    Line-by-line comments**

Line 56 "has also studied the relationship between debris-covered glaciers and rock glaciers."

Please add 'transient' between 'the' and 'relationship' otherwise the way this is written makes it seem like the

model is only being used in a steady state sense from the text immediately before.

Added.

Line 109: "Østrem curve data representative of a medium-sized Alpine glacier."

Consider revising this justificatoin as it is unclear what the size of a glacier has to do with the shape of Østrem?s curve.

We changed the text to "The parameter $D_0$ is chosen based on an Østrem curve that is representative for data from Zmuttgletscher, a medium-sized Alpine glacier."

Line 126: "In addition, we accounted for the fact that the terminal ice velocity in SIA goes to zero, which is not physically realistic, by taking an averaged velocity over several ice thicknesses near the terminus when computing the debris transport."

How does the ice not go to zero velocity at the toe on a real glacier? If ice thickness is 0 velocity must be zero. How physically does ice flow remove debris from a glacier? This is not the correct process for the removal of debris at the toe, which can only leave by ice cliff retreat or by geomorphic trundling down the glacier slope.

We changed the text to "In addition, we accounted for the fact that the ice velocity in SIA goes to zero at a faster rate near the terminus than is physically realistic by taking an averaged velocity over several ice thicknesses near the terminus when computing the debris transport."

Line 130-132. "We note that although the terminal boundary condition used here is similar to the one implemented in Anderson and Anderson (2016), since in both cases the glacier ends with a small debris-free terminal ice cliff, our choice of boundary condition has the subtle but important feature of being independent of grid size."

I see in the appendix that the terminus formulation presented in this manuscript is in fact grid size dependent.

It seems we were not clear enough in the text with regard grid size dependency. Our formulation of the boundary condition is strictly speaking grid size independent, but only slightly for the small grid sizes used in the manuscript. Although the resulting glacier extent does vary with grid size, it converges to a fixed length for decreasing grid sizes as indicated further below in the response. We adjusted to text accordingly in the text (Methods, end of section 2.1.4. and in Appendix A) and also added this convergence (in Table A1 in Appendix A and in figure S6 in the supplementary material).

In section 2.1.4 we changed the text to: "We note that our boundary condition is similar to the one implemented in Anderson and Anderson (2016), but our approach differs in that we remove debris beyond a critical thickness (position of ice cliff), whereas in Anderson and Anderson (2016) it is removed from a terminal wedge. Although our ice cliff position is by construction not really grid size dependent, the modelled terminus position shows some dependency on grid size. However, sensitivity tests demonstrate that there is fast convergence with decreasing grid size and the dependency for the 25 m grid size resolution used here essentially vanishes (see Appendix A, Table A1 and supplementary figure A2).")

Line 132-133: "Further details of the boundary condition, including the interpolation scheme between grid points and convergence tests for different grid sizes, is found in Appendix A." I did not find a reproducible explanation of the interpolation scheme and how it differs from A & A, 2016 nor the convergence tests for different grid sizes in Appendix A. Please update accordingly.

Thank you, this material has been added in Appendix A. Please see below for further details.

Figure 1. Nice addition!

Thank you.

Line 174. Helpful explanation of what is to come.

Thank you.

Line 354. "We showed that the memory of a debris-covered glacier is selective, exhibiting an effective hysteresis, with periods of relatively cold climate having a sustained effect on the volume and in particular on the length."

While this is a clear result from the modeling and model set up (Fig 6), I am concerned about the effect of an assumed constant englacial debris concentration through time on this conclusion.

Perhaps one way around this issue is to state that the effect you see with glacier size enhancement at cooler temps and glacier shrinking at warmer temps is consistent with the englacial debris concentration changing in time but that it appears to be independent. I am not sure what simulations can be done with this model to show that this hysteresis is in fact not model assumption dependent.

We have addressed this concern in the response to the general comments above. In addition, in the Model limitations we are quite clear about the shortcomings of the model with regard to variable debris input, but we added a few things in the text to make it even clearer (see in main response for details). And again the additional model experiment with a more sophisticated model (not included in the manuscript since it is out of scope) that ensures constant debris input really does confirm our conclusion from the simpler model.

Line 357. "Previous numerical simulations of the transient response of debris-covered glaciers focused only on the effects of sudden debris input in the form of an avalanche (Vacco et al., 2010; Menounos et al., 2013). Such a one-time debris input leads to an advance in glacier extent and foreshadows the results of our study, where a constant debris source and changing climate forcing gives rise to a more complex response."

This is statement is not accurate and omits a number of other studies of the transient modeling of DCGs in response to climate change.

Other studies have modeled the transient response of DCGs to climate change with a steady input of debris. Not only a one time avalanche of debris. Please the Rowan et al. (2015) model of Khumbu glacier responding to a warming climate. See Anderson et al., 2018 (Glaciation of alpine valleys: The glacier-debris-covered glaciers continuum) and Crump et al., 2017 for examples of modeled glaciers that contradict this above statement.

Studies that show transient response of debris-covered glaciers to climate change:

Rowan et al. (2015). The whole paper focuses on transient responses to climate change with a steady debris deposition rate.

Anderson et al., 2018: See figures 18 and 19 (for a single climate cycle) , and figure 20 for the transient change in a debris-covered glacier over 7 climate cycles.

Crump et al., 2017: See Figures 6 and 7 show a debris covered glacier responding to step changes in ELA over a 4000 year period.

The statements in the text here need to be adjusted to reflect the contributions of these previous efforts.

We did not mean to imply that these were the only studies that involved transient response of debris-covered glaciers. All of these papers are already cited in the introduction. Rather, we wished to highlight these two studies as they relate to the topic of debris-covered memory. To clarify this, we have added the word "some" at the beginning of the sentence. We have also added the citation of Crump et al. (2017) at the end of the section, as it is relevant here.

Line 363. "This also means that for debris covered glaciers, no unique glacier length exists for a given climate, but rather that the length of debris covered glaciers is determined by the history of repeated cold phases. Furthermore, debris-covered glaciers under random climate forcing are expected to have a longer average length than the steady state length corresponding to the equivalent constant climate forcing."

See comments above.

As noted in the response to the reviewer's major comments and the comments to Line 354, we have already addressed this issue above in the response to the main comments. We note however here again that we clarified in the text that the tendency of advance/or stangation in length refers to century climate fluctuations and that for much longer time scales, retreat to warming occurs in our modelling as well.

Line 439. Maybe should be up to 12% here based on Anderson et al., (2021).

"Kennicott Glacier exhibits the highest fractional area of ice cliffs (11.7 %) documented to date."

Thank you, this has been corrected.

Line 476. "Equation (14) is similar to an equation derived by Anderson and Anderson (2018) but with the important difference that we have allowed variable velocity in this derivation, rather than assuming a constant velocity along the entire glacier."

Which equation are you referring to here from A & A (2018)?

There are two models presented and discussed in A & A (2018). And variable velocities are explicitly discussed and modeled in that paper. So I am confused as to what you are referring to. Please clarify in the text as it creates a false sense that spatially variable velocities were not considered in A & A (2018).

We are referring to Equation (27) of Anderson and Anderson (2018). The entire analytical derivation depends on the assumtion of constant velocity, e.g. "For the analytical model, we assume a uniform $U_s$ field in the debris-covered portion of the glacier so the right-hand term in Eq. (9) is neglected." While variable velocities are considered in the numerical studies performed, the analytical argument we are referred to does not do so. This is in contrast to our argument, which holds true for variable velocity. Just so there is no confusion here, we have added the equation number from Anderson and Anderson (2018) in the text.

**Appendix A**

See notes in the main comments above.

Line 575. "A similar boundary condition was used in Anderson and Anderson (2016), where a debris- free region was also employed at the terminus. In their approach, the size of the terminal ice cliff depends directly on the grid spacing but the debris flux can be varied. In contrast, our terminal ice cliff is roughly independent of grid size, since it depends on the critical ice thickness H* , but the debris flux is fixed by the local ice velocity."

This is not correct and should be adjusted. The ice cliff height in A & A 2016 does not vary depending on grid size and the ice cliff height is fixed. There is no debris-free region on the modeled glacier. The grid size dependence is explained in A & A (2016):

"In our model, the debris thickness h debris (x, t) represents a layer of equal thickness on any cell. Debris thickens more slowly with a larger dx because the debris volume advected into a cell is spread over a larger area (due to the larger dx; dy=1; dy (in m)). There is therefore a timescale built into the thickening of debris in a cell that is dependent on dx. Because ablation rates are sensitive to debris-cover thickness, changing dx has an effect on glacier evolution."

This effect is the same in this model as well. As dx is decreased the effect of dx on glacier evolution is reduced. It seems that the model of A &A (2016) and the model presented here have the same issue, namely that there is grid size dependence in the length that converges as dx is reduced. Please update the text to accurately represent the similar issues for the models as right now the text states that the model presented here is grid size independent when it is not actually.

We have rewritten Appendix A to address the reviewer's comments both here and in the Major comments section. In particular, we have clarified the differences between our boundary condition and the boundary condition used in Anderson and Anderson (2016) and we have also included convergence tests. We note that the reviewer is not correct when he states that "the ice cliff height in A & A 2016 does not vary depending on grid size and the ice cliff height is fixed". The ice cliff geometry, which consists of the entire terminal wedge used in Anderson and Anderson (2016), depends on the grid size due to two reasons: 1) the terminal wedge height depends on the glacier thickness at this point, with a larger grid spacing necessarily meaning a larger wedge height; 2) the terminal wedge length also depends directly on grid size, since it is equal to the grid size plus whatever length remains to complete the terminal wedge (see Fig. A1 in Anderson and Anderson (2016)).

This formulation for the terminal wedge is potentially problematic because it is here that the boundary condition is applied. If the size of this region changes with the discretization, then this is likely to cause convergence issues. For example, the boundary will get smaller and closer to the terminus as the grid size is reduced, requiring that an increased amount of debris is squeezed off the glacier at a faster rate from a smaller region per unit time. It is unclear whether such a formulation would converge to a stable steady state. This issue can be avoided by designing a boundary condition that represents the same physics and geometry in a grid size invariant way, which is the case for the boundary condition used in this manuscript.

We agree with the reviewer that the method employed in our manuscript exhibits a grid size dependence and it is important to be transparent about this. In particular, the steady state length varies with the choice of grid size. We have shown that despite this, the effect is not large and the glacier length converges as the grid size is reduced. The grid size used in the present study, dx = 25 m, at the debris concentration of $c = 0.025\%$ compares favourably with the smallest grid size used in the convergence testing, with an error in absolute length of roughly 200 m or about 5% of the total length. As the reviewer points out, this grid size dependency may be an inherent feature of numerical debris-covered glacier models. We agree with this point.

However, the reviewer's claim that the model of Anderson and Anderson (2016) has a similar convergence behaviour has not been substantiated. No convergence testing was performed in the original paper, as instead of studying the behaviour of the model for successively smaller values of dx to see if it tends to a limiting value, the authors reported on one test with a larger value of dx. As such, we feel that the boundary condition used in our study represents an important step forward.

Here is the rewritten text from Appendix A:

The value of $x^*$ is found using a sub-grid linear interpolation on the grid points $x_j$ and the corresponding ice thicknesses at these grid points, $H_j = H(x_j)$. Hence $x^*$ is bounded by the grid points $x_i$ and $x_{i+1}$ such that $H(x_i) > H^*$ and $H(x_{i+1}) < H^*$, as shown in Fig A2. All the surface debris transported past the ice cliff location $x^*$ slides down the cliff and out of the system and is therefore removed from the surface by setting the debris thickness to zero for all grid points past $x^*$. The surface mass balance calculation at $x_{i+1}$ accounts for the sub-grid location of the ice cliff by using a weighted average of debris-covered and bare ice melt rates, with the weighting dependent on location of $x^*$, and is given by:

$$a_{i+1} = a \frac{x^* - x_i}{\Delta x} + \tilde{a} \frac{x_{i+1} - x^*}{\Delta x}, \tag{A1}$$

where $a$ is the debris-covered surface mass balance, $\tilde{a}$ is the debris-free surface mass balance, and $\Delta x$ is the grid spacing.

Although the overall method is grid size dependent, it is convergent, i.e. the steady state glacier extent converges to a fixed value as the grid size is reduced. The results of convergence tests for the case of $c = 0.025\%$ for both warm, corresponding to ELA = 3100 m, and cold, corresponding to ELA = 3000 m, are shown in Table A1. The extents for the smallest grid size $\Delta x = 6.25$ m are used for the relative error calculations. We note that the relative error in steady state glacier extent for the grid size used throughout the present study is roughly 5%, which is on the order of about 200 m. A plot of steady state glacier profiles corresponding to the tests shown in Table A1 can be found in the supplementary material.

A similar boundary condition was used in Anderson and Anderson (2016), where an ice cliff is also employed

Table 1: Convergence test results for steady state glacier extent corresponding to $c = 0.025\%$.

| $\Delta x$ (m) | Extent (warm) (km) | Rel. error | Extent (cold) (km) | Rel. error |
|---|---|---|---|---|
| 100 | 5.86 | 0.281 | 9.66 | 0.241 |
| 50 | 6.97 | 0.145 | 11.17 | 0.122 |
| 25 | 7.71 | 0.055 | 12.13 | 0.046 |
| 12.5 | 7.99 | 0.021 | 12.48 | 0.019 |
| 6.25 | 8.16 | – | 12.72 | – |

at the terminus and similar physics governs debris leaving the system, due to either tumbling down the terminal cliff or else by cliff backwasting. One difference in their approach compared with ours is that debris covers the glacier right up until the terminus and leaves the system at the ice cliff at a prescribed rate, which for most simulations is set equal to the melt rate times the debris thickness. A second difference is the fact that the size of the terminal ice cliff – and hence the size of the region over which the terminal debris flux condition is applied – depends directly on the grid spacing, resulting in a smaller ice cliff as the grid size is reduced. The implications of this for model convergence are not clear.

In contrast, the geometry terminal ice cliff used in the present study is essentially independent of grid size, since it depends on the critical ice thickness $H^*$. As the grid size is reduced, the position of the ice cliff is more accurately determined and therefore the terminus geometry converges to a steady value.

And here is the additional figure, now part of the supplementary material, where panel (a) represents the steady state profiles for ELA = 3100 m and (b) represents the steady state profiles for ELA = 3000 m, in both cases with $c = 0.0025\%$:

[revised manuscript text omitted]

---

## Author Response (AR3)

**Response letter to review for "Modelling steady states and the transient response of debris-covered glaciers" by J.Ferguson and A.Vieli**

Dear Dr. Zekollari,

Thank you for continuing to handle the review process for our manuscript. We have addressed both of the reviewer's comments below.

Best wishes,
James Ferguson and Andreas Vieli

**1 Summary**

We would like to thank the reviewer for a final helpful review. In order to address the clarifications and suggestions, we have reworded parts of the manuscript. In what follows, we show first the reviewer's comments in black and then our response in blue. At the end of this document, we provide a marked-up version of the manuscript which shows the changes that were made between the original and the newly revised.

**2 Reviewer comments**

Thank you for your thorough replies. I appreciate you taking the time to clarify the issues raised.

There are only couple of small items that should be clarified/revised:

Line 489-490.

"Equation (14) is similar to equation (27) derived by Anderson and Anderson (2018) but with the important difference that we have allowed variable velocity in this derivation, rather than assuming a constant velocity along the entire glacier."

Equation (27) in A and A, 2018 does not assume uniform velocity for any part of the glacier, rather it only assumes that englacial debris emergence is negligible. It is derived directly from the continuity equation for debris and is independent from the analytical model presented in that paper (A and A, 2018). Please revise the text above to reflect the fact that variable surface velocities are not assumed in the derivation of equation (27) from A and A, 2018.

The text from A and A, 2018 is below for completeness:

**4.3. Concave up debris patterns: velocity controlled**

Debris emergence is negligible if there is no debris within the glacier or if sub-debris melt rates are small. In such cases Eq. (9) reduces to

$$\frac{\partial h_{debris}}{\partial t} = \frac{\partial (h_{debris} U_s)}{\partial x} \qquad (26)$$

Solving for $h_{debris}$ by taking the integral of Eq. (10) with respect to $x$ leads to

$$h_{debris}(x) = \frac{q_e}{U_s} \qquad (27)$$

where $q_e$ is the surface parallel flux of debris at the down-glacier end of the emergence zone (where the emergence rate declines to zero; see Glazyrin, 1975). Here $q_e$ is constant with respect to $x$ and is the integral of emergence rate. Hence

$$q_e = \int\limits_{x_e}^{x_{e_{end}}} \frac{Cb'}{(1-\phi)\rho_r} dx \qquad (28)$$

And equation 9 from A and A, 2018:

In the ablation zone, the rate of change of surface debris thickness in one-dimension may be written

$$\frac{\partial h_{debris}}{\partial t} = \frac{Cb'}{(1-\phi)\rho_r} - \frac{\partial (U_s h_{debris})}{\partial x} \qquad (9)$$

where $b'$ is the sub-debris ice melt rate, $U_s$ is the down-glacier surface speed, and $\rho_r$ is the density of rocks comprising the debris cover (e.g., Nakawo et al., 1986; Naito et al., 2000). The first term on the right-hand side is a local debris emergence rate $\varepsilon_{debris}$ (in units of m y$^{-1}$), and the second represents the thickening or thinning rate of debris caused by down-valley gradients in the surface debris discharge $U_s h_{debris}$. For the analytical model, we assume a uniform $U_s$ field in the debris-covered portion of the glacier so the right-hand term in Eq. (9) is neglected.

Thank you for this clarification. We have changed the text to read:

"Equation (14) is similar to equation (27) derived by Anderson and Anderson (2018) but with the difference that we have not assumed negligible englacial debris emergence and that our equation is only applied at the terminus."

Line 602.

"A second difference is the fact that the size of the terminal ice cliff – and hence the size of the region over which the terminal debris flux condition is applied – depends directly on the grid spacing, resulting in a smaller ice cliff as the grid size is reduced. The implications of this for model convergence are not clear."

Ice cliffs are defined quite differently in the two papers/terminal parameterizations.

The terminal wedge parameterization in A and A, 2016 prevents the issue outlined by the authors above for the grid sizes explored there (100 and 200m). The ice cliff height is set to 10 m, is vertical, and is applied within the terminal wedge. Here is the ice cliff parameterization as done in A and A, 2016:

[Figure]

The ice cliff height does not change in A and A, 2016 because it is defined as a parameter in the code and the ice cliff height is always much smaller than the ice thickness at the start of the terminal wedge. The code from A and A, 2016 is not yet available (it should be and I would like to make it so) so this is not clear to the authors here.

I would simply appreciate if the text quoted above was at least edited to reflect that the ice cliff is defined differently between the papers and as it is defined in A and A, 2016 the ice cliff height is not grid size dependent.

It is accurate though to say that the height of the start the terminal wedge is grid sized dependent. But because the terminal wedge is debris covered in A and A, 2016 it does not introduce ice cliff height dependence issues as described by the authors.

Thank you for the clarification. This is helpful to understand the boundary condition, since these details are not clearly reported in Anderson and Anderson (2016). For example, we could not find any mention of a fixed ice cliff height in the text and the figure sketched by the reviewer does not match Figure A1 from the paper (with no mention of ice cliff height), shown here for comparison:

[Figure]

There is still a potential issue with a fixed ice cliff height as reported by the reviewer. This is because as the grid spacing is decreased, it could result in a terminal wedge height greater than the ice cliff height (found at the terminus), with unclear consequences for convergence. Perhaps as the reviewer suggests, this is not an issue, though it has not been demonstrated in convergence tests. In any case, this outside the scope of our paper.

Although we are not able to refer in our paper to unpublished reviewer comments, we have reworded text in

the following way, to more accurately reflect how we now understand the boundary condition to work:

[revised manuscript text omitted]